

# Aerosol Optical Extinction during the Front Range Air Pollution and Photochemistry Éxperiment (FRAPPÉ) 2014 Summertime Field Campaign, Colorado U.S.A.

Justin H. Dingle[1], Kennedy Vu[1], Roya Bahreini[2], Eric C. Apel [3], Teresa L. Campos[3], Frank Flocke[3], Alan
Fried[4], Scott Herndon[5], Alan J. Hills [3], Rebecca S. Hornbrook [3], Greg Huey [6], Lisa Kaser [3], Denise D.
Montzka[3], John B. Nowak[5], Mike Reeves[3], Dirk Richter[4], Joseph R. Roscioli[5], Stephen Shertz[3], Meghan
Stell[3], David Tanner[6], Geoff Tyndall[3], James Walega [4], Petter Weibring[4], Andrew Weinheimer[3]

[1] Environmental Toxicology Graduate Program, University of California, Riverside, CA 92521
[2] Department of Environmental Sciences, University of California, Riverside, CA 92521
[3] National Center for Atmospheric Research, Boulder, CO 80301
[4] Institute for Arctic and Alpine Research, University of Colorado, Boulder, CO 80303
[5] Aerodyne Research, Inc., Billerica, MA 01821
[6] Department of Earth and Atmospheric Sciences, Georgia Institute of Technology, Atlanta, GA 30033

*Corresponding author*: R. Bahreini (roya.bahreini@ucr.edu)

**Abstract.** Summertime aerosol optical extinction ($\beta_{ext}$) was measured in the Colorado Front Range and Denver Metropolitan Area as part of the Front Range Air Pollution and Photochemistry Experiment (FRAPPÉ) campaign during July-August 2014. An Aerodyne Cavity Attenuated Phase Shift particle light extinction monitor (CAPS-PM$_{ex}$) was deployed to measure dry, $\beta_{ext}$ of submicron aerosols at $\lambda=632$ nm at 1 Hz. Data from a suite of gas-phase instrumentation were used to interpret $\beta_{ext}$ behavior under various categories of air masses and sources. Extinction enhancement ratios relative to CO ($\Delta\beta_{ext}/\Delta CO$) were significantly increased in highly aged air masses compared to fresh air masses by 50-60%. The resulting increase in $\Delta\beta_{ext}/\Delta CO$ under highly aged air masses was accompanied by formation of secondary organic aerosols (SOA). In addition, the impacts of aerosol composition on $\beta_{ext}$ in air masses under the influence of urban, natural oil and gas operations (O&G), and agriculture and livestock operations were evaluated. Estimated non-refractory mass extinction efficiency (MEE) values for different air mass types ranged from 1.83-3.30 m$^2$ g$^{-1}$, with the minimum and maximum values observed in agriculture and urban + O&G influenced air masses, respectively. The mass distribution for organic, nitrate, and sulfate aerosols presented distinct profiles in different air mass types. During Aug. 11-12, regional influence of a biomass burning event was observed, increasing the background $\beta_{ext}$ by 10-15 Mm$^{-1}$ and the estimated MEE and $\Delta\beta_{ext}/\Delta CO$ values in the Front Range.





## 1 Introduction

Aerosol optical extinction coefficient ($\beta_{ext}$) represents the attenuation of light due to aerosol absorption and scattering of solar radiation. For a population of aerosol particles, $\beta_{ext}$ depends on aerosol size, composition, particle number concentration, shape and morphology (Bohren et al. 1998). Atmospheric aerosols have important implications on climate. They modify the Earth's radiative energy budget directly through absorption and scattering of light, and indirectly through changing cloud characteristics (e.g., cloud droplet number concentration, cloud droplet size, cloud reflectivity, or lifetime) (Ramanathan et al. 2001, Seinfeld et al. 2006, Langridge et al. 2011). In addition, aerosols with diameters between 0.1 μm to 1 μm are the main contributors to visibility degradation in anthropogenically polluted areas and on regional scales due to their direct interactions with solar radiation (Malm 1989, Hobbs 2000, Ying et al. 2004). For example, it has been observed that the important contributors to light scattering in the Colorado Rocky Mountains are particulate matter from the urban emissions (Levin et al. 2009). The Denver Metropolitan area has also experienced seasonal air pollution and visibility degradation in the past. The wintertime pollution in Denver when trapped closer to the surface due to the low inversion layer causes a greyish-brown cloud referenced to as "Denver Brown Cloud." The composition of the Denver Brown Cloud and contribution of different chemical species to the observed $\beta_{ext}$ during the wintertime have been investigated in 1970's to late 1980's (Groblicki et al. 1981, Wolff et al. 1981, Neff 1997). These studies concluded that among all the measured aerosol species, elemental carbon, ammonium sulfate, and ammonium nitrate contributed the most (37.7%, 20.2%, and 17.2 %, respectively) to optical extinction in the visible range. Despite the extensive studies of the Denver Brown Cloud in 1970s and 1980s, recent comprehensive summertime characterization of air quality in the Colorado Front Range has been lacking.

The meteorological influence on air quality and visibility in the Front Range can also be important. Typically during the day, easterly upslope flow transports emissions from local sources westward while during the night, the flow reverses and downslope drainage flow through Platte River Valley sets in. Occasionally, a synoptic scale cyclone, called the Denver Cyclone, is established when drainage flow of air masses is prevented due to propagation of a vortex that develops east of the Rocky Mountains, contributing to transport and mixing of air masses in a cyclonic flow pattern (Crook et al. 1990, Reddy et al. 1995).

With a twofold increase in natural resource extraction since 2005 to about 24,000 active oil and natural gas (O&G) production wells in 2012, northeastern Colorado has experienced extensive fossil fuel production within the past decade (Scamehorn 2002, Pétron et al. 2014). Emissions from O&G operations are not limited to methane and non-methane hydrocarbons, but also include emissions from industrial equipment known to emit other volatile organic compounds (VOCs), nitrogen oxides, and particulate matter (Gilman et al. 2013). One of the major air quality issues the Colorado Front Range faces is the exceedance of the 8-hour National Ambient Air Quality Standard (NAAQS) standard for ozone (75 ppbv) during the summertime. In 2007, the Denver metropolitan area and the northern parts of the Colorado Front Range were classified as nonattainment areas due to the summertime elevated levels of ozone (www.colorado.gov/cdphe/attainment). In the wintertime, it has been estimated that ~55% of the VOC OH-reactivity was attributed to O&G operations, indicating that photochemical



production of ozone is significantly impacted by O&G emissions (Gilman et al. 2013). Summertime impacts of such emissions on the formation of secondary pollutants and visibility reduction in the Front Range have not been explored previously. In addition to local point and area sources in the Front Range, biomass burning emissions from wildfires in the region may also act as a source of aerosols, contributing to regional haze (Park et al. 2003).

5        During July-August 2014, airborne measurements were conducted over the Colorado Front Range as part of the Front Range Air Pollution and Photochemistry Éxperiment (FRAPPÉ) to characterize the influence of sources, photochemical processing, and transport on atmospheric gaseous and aerosol pollutants in the area. This paper will discuss the role of local aerosol sources in the Front Range and regional wildfires on aerosol optical extinction in the absence of the Denver Cyclone by investigating chemical and optical properties of aerosols in different air masses.

10 **2 Measurements**

**2.1 FRAPPÉ Field Campaign**

*In-situ* measurements were conducted aboard the National Science Foundation/National Center for Atmospheric Research (NSF/NCAR) C-130 aircraft during July 20-August 18, 2014. Flight tracks of the C-130, color coded with different trace gases, are presented in Figure 1a-c. In the current analysis, airborne data were limited to those obtained only in the boundary layer of 15 the Colorado Front Range (i.e., altitudes below 2500 m east of the foothills and below 5500 m with easterly winds over the foothills and the Continental Divide) to capture the influence of local sources such as power plants, O&G, agriculture, livestock, and urban emissions. Occasionally, when air masses with the mountain-valley circulation patterns were sampled, data from higher altitudes (< 4000 m) over the Denver Metropolitan area were also considered.

**2.2 Instrumentation and Methodology**

20 The NSF/NCAR C-130 aircraft carried an extensive collection of instruments for the characterization of the diverse atmospheric pollutants in the Colorado Front Range. The relevant instrumentations used in the current analysis are described below. (The data produced by these instruments are available at http://www-air.larc.nasa.gov/cgi-bin/ArcView/discover-aq.co-2014?C130=1).

       The parameter $\beta_{ext}$ at 632 nm was measured using a Cavity Attenuated Phase Shift particle light extinction monitor 25 (CAPS-PM$_{ex}$) (Aerodyne Research Inc., Billerica, MA). The CAPS-PM$_{ex}$, utilizes high reflectivity mirrors at two ends of a 26-cm long, near-confocal cavity. Under the particle free sampling mode, the LED light output is directed to the first mirror, then through the second mirror to the photodiode detector, producing a slightly distorted waveform, whereas under the aerosol sampling mode, the detector detects a greater distorted waveform, characterized by a phase shift. The observed phase shift is then related to aerosol $\beta_{ext}$ as follows:

30

$$\beta_{ext} = (\cot\theta - \cot\theta_o) * (2\pi f/c), \tag{1}$$





where $\cot\theta_o$ is the phase shift from the particle-free sampling mode, $\cot\theta$ is the phase shift during the aerosol sampling mode, f is the frequency, and c is the speed of light. The estimated uncertainty in $\beta_{ext}$ is 10%, while the 3-$\sigma$ detection limit under particle free air is 3 $Mm^{-1}$ (Massoli et al. 2010, Petzold et al. 2013). Measurements of the baseline were conducted through the system's internal filter unit regularly, at 5 minute intervals. The filter period, which lasted for 1 minute, included 10 s of flush time, 40 s of filter sampling, followed by another 10 s of flush time. When the baseline values shifted by more than 5 $Mm^{-1}$, the baseline values were interpolated for a more accurate estimation of optical extinction. The measured $\beta_{ext}$ includes the combined effects of scattering and absorption of light by aerosols at 632 nm, given relatively high single scattering albedo values of aerosols downwind of urban environments (Langridge et al. 2012), $\beta_{ext}$ is expected to be dominated by scattering coefficient. As discussed in Section 3.4, in biomass burning-influence air masses, contribution of absorption by black carbon to $\beta_{ext}$ could be more significant.

The CAPS-PM$_{ex}$ shared a common inlet with a compact Aerosol Mass Spectrometer (mAMS; Aerodyne Inc., Billerica, MA) coupled with a time-of-flight (TOFwerk, Thun, Switzerland) detector to measure particle mass distribution and non-refractory submicron aerosol composition (NR-PM$_1$) of organics, nitrate, sulfate, chloride, and ammonium (Jayne et al. 2000, Drewnick 2005, Canagaratna et al. 2007). The estimated uncertainty for all aerosol species is 30% (Bahreini et al. 2009). Both instruments sampled particles through a secondary diffuser mounted inside a NCAR HIAPER modular inlet (HIMIL), mounted facing forward, under the C-130 aircraft. Given the total flow rate within the inlet and assuming particle density of 1500 $kg/m^3$, ambient temperature of 20 °C, and ambient pressure of 70 KPa, 2 µm particles were estimated to be transmitted by 50%, making the inlet nominally a PM$_2$ inlet. Residence time in the inlet was estimated to be ~0.56 s. The CAPS PM$_{ex}$ was set to measure at 1 Hz frequency. Reported $\beta_{ext}$ values were normalized for STP (273 K and 1 atm) conditions.

The relationship between the primary emitted nitrogen oxides (NO$_x$) and the higher oxidized species of nitrogen intends to capture the transformation of NO$_x$ in the atmosphere upon aging (Kleinman et al. 2007, Langridge et al. 2012). Thus, measurements of nitric oxide (NO), nitrogen dioxide (NO$_2$), nitric acid (HNO$_3$), alkyl nitrates (ANs), peroxy acetyl nitrate (PAN), and peroxy propionyl nitrate (PPN) were used to calculate the ratio of primary nitrogen oxides (NO$_x$ = NO + NO$_2$) to NO$_y$ (NO$_y$ = NO$_x$ + HNO$_3$ + ANs + PAN +PPN) in order to track the extent of photochemical aging in an air mass with non-zero emissions of NO$_x$ (Kleinman et al. 2007, DeCarlo et al. 2010). A ratio that yields a value close to one represents air masses that are relatively fresh whereas a ratio closer to zero represents more aged air mass. NO and NO$_2$ were measured using the NCAR 2-channel chemiluminescence instrument (Ridley et al. 1990). A chemical ionization mass spectrometer (CIMS) coupled with a quadrupole detector was operated to measure HNO$_3$, using SF$_5$$^-$ as the reagent ion (Huey et al. 1998, Huey 2007). ANs were measured using thermal dissociation-laser induced fluorescence (TD-LIF) (Thornton et al. 2000, Day et al. 2002). PAN and PPN species were measured using the NCAR PAN-CIGAR CIMS (Zheng et al. 2011), with I$^-$ as the reagent ion.

The impacts of different pollution sources on sampled air masses were characterized by considering several auxiliary gas-phase tracers. Carbon monoxide, the tracer for combustion emissions, was measured by VUV-fluorescence with the





estimated uncertainty of 3% (Gerbig et al. 1999). Ethane ($C_2H_6$), used to identify influence of O&G emissions, was measured using the Compact Atmospheric Multi-species Spectrometer (CAMS), employing infrared spectrometry (Richter et al. 2015). The Aerodyne dual quantum cascade trace gas monitor for ammonia ($NH_3$) equipped with a mid-infrared absorption spectrometer was used to identify emissions of agriculture and livestock operations (Ellis et al. 2010). The influence of biomass

burning was identified using the measurements of hydrogen cyanide from the NCAR Trace Organic Gas Analyzer (TOGA), a fast gas chromatography coupled with a quadrupole mass spectrometer (GC-MS) set to selected ion monitoring mode for quantification (Apel et al. 2015) and acetonitrile by proton-transfer reaction mass spectrometry (PTR-MS), a high sensitivity instrument with fast time response that employs a quadrupole mass spectrometer to measure volatile organic compounds (de Gouw et al. 2007, Karl et al. 2009). A Passive Cavity Aerosol Spectrometer Probe (PCASP) instrument which measured

aerosol size distributions under ambient conditions in the size range of 0.1-3 μm was utilized to determine the particle number concentrations under different sources or air mass types as described above (Rosenberg et al. 2012).

## 3. Results and Discussion

### 3.1 Aerosol optical extinction characterization under different photochemical aging regimes

$NO_x/NO_y$ ratios were observed to be highest over the city representing freshly emitted plumes from vehicular traffic (Figure

S1). Away from the city center, $NO_x/NO_y$ values decrease, representing the relative evolution of fresh air masses containing $NO_x$. Figure 2 shows the scatter plot of $\beta_{ext}$ vs. CO, color coded with the $NO_x/NO_y$ ratio, on July 26, 29, 31 and August 2-3, 7-8, 15-16, 18 (i.e., excluding days with the influences of the Denver Cyclone and biomass burning events). The extinction enhancement ratios $\Delta\beta_{ext}/\Delta CO$ under 3 aging regimes, categorized by $NO_x/NO_y$ ratio values, were analyzed by weighted linear orthogonal distance regression (ODR) fits, with the slopes representing the enhancement ratios. In these fits, weights

represented standard deviations equal to the uncertainties in CO (3%) and $\beta_{ext}$ (10%). $NO_x/NO_y$ values of <0.3, 0.3-0.7, and > 0.7 represent relatively aged, intermediately aged, and fresh $NO_x$-containing air masses, respectively. Measurements here focused on air masses impacted by all sources. Different trends in $\Delta\beta_{ext}/\Delta CO$ were seen under the three aging regimes, with the lowest values of 0.16 ± 0.005 Mm$^{-1}$ ppbv$^{-1}$ and 0.15 ± 0.004 Mm$^{-1}$ ppbv$^{-1}$ observed in the fresh and intermediately aged air masses, respectively. On the other hand, the most aged air masses showed the highest $\Delta\beta_{ext}/\Delta CO$ value of 0.25 ± 0.004 Mm$^{-1}$

ppbv$^{-1}$, indicating a factor of 1.6 increase in the extinction enhancement ratio due to photochemical aging. The correlation coefficient r values were 0.63, 0.73, and 0.64 for fresh, intermediately aged and aged air masses, respectively. Note that the $NO_x/NO_y$ photochemical clock provides a true measure of atmospheric processing only for air masses influenced by emissions from high-temperature combustion processes, e.g., on-road or off-road vehicular exhaust. Therefore, age classification based on the $NO_x/NO_y$ value does not differentiate between a purely combustion-derived air mass with a certain photochemical age

and the same air mass mixed with a fresh or aged plume from non-combustion sources. Therefore, it is possible that the plumes categorized with a given $NO_x/NO_y$ value were non-uniformly mixed with differently aged, non-combustion influenced air masses, resulting in a lower than optimum correlation coefficients in Figure 2. Similarly, a photochemical clock based on the

ratios of different VOCs does not provide an accurate estimate of plume processing times in this environment due to different emission ratios of most VOCs from urban and O&G sources. Regardless of this caveat, the bulk of the organic aerosol mass during the daytime in the Front Range was characterized as SOA due to the observed significant increase in the enhancement ratio of OA to CO with aging: $\Delta OA/\Delta CO$ was a factor of ~ 3 higher in air masses with $NO_x/NO_y <0.3$ (0.073 ± 0.002 µg m$^{-3}$

ppbv$^{-1}$, r=0.76) compared to those with $NO_x/NO_y >0.7$ (0.027 ± 0.004 µg m$^{-3}$ ppbv$^{-1}$, r=0.67). However, the $\Delta NO_3^-/\Delta CO$ and $\Delta SO_4^{2-}/\Delta CO$ enhancement ratios did not increase with photochemical aging and demonstrated poor overall correlation coefficients (r <0.27 for $\Delta NO_3^-/\Delta CO$ and r <0.19 for $\Delta SO_4^{2-}/\Delta CO$). Therefore, the increase in the enhancement ratio of aerosol optical extinction coefficient with CO was likely also driven by SOA formation.

## 3.2 Impacts of source and aerosol composition on aerosol optical extinction

Analysis of the average composition of NR-PM$_1$ in the Northern Front Range, in the absence of the Denver Cyclone, revealed significantly higher concentrations of organic aerosols relative to inorganic anions in the urban- and urban + O&G-influenced air masses, with a fractional contribution of ~74% (Figure 3). In O&G and agriculturally influenced air masses, organic fraction was lower, at 65% and 57%, respectively. On average, similar concentrations of non-refractory aerosol sulfate and chloride were observed in the different air masses while concentration of nitrate aerosols increased by a factor of ~2-3 in agriculturally-

influenced air masses compared to the other air mass types.

Aerosol optical extinction values under the influence of different sources were further analyzed using auxiliary gas-phase data. CO, $C_2H_6$, and $NH_3$ tracers represent urban emissions, O&G, and agricultural and livestock operations, respectively. Urban emissions were classified with enhancement in CO relative to the background (95-105 ppbv), while O&G and agricultural emissions were classified using enhancements of $C_2H_6$ over 2500-3000 pptv, and that of ammonia over 5 ppbv,

respectively. A fourth air mass classification used in this analysis, urban + O&G, is the combination of the urban and O&G air masses when both CO and $C_2H_6$ mixing ratios were elevated above the background. The background mixing ratios for each gas tracer were determined by the mode of the frequency distribution of the tracer's mixing ratio observed in each flight. The impacts of sources and aerosol composition on extinction were explored by considering correlation coefficients of least-squared regression fits to the scatter plots of aerosol extinction vs. the mass concentration of the three dominant aerosol species

(OA, nitrate aerosols, and sulfate aerosols) in urban-, O&G-, and agricultural-influenced air masses.

Figure 4 shows the correlation coefficient (r) values of dry extinction vs. aerosol species mass concentration, in different air mass types as characterized above. The relationship between dry $\beta_{ext}$ vs. OA revealed a strong positive correlation under urban, O&G, and urban + O&G air masses (r = 0.55, 0.71, 0.55, respectively). This observation, combined with the evolution of $\Delta\beta_{ext}/\Delta CO$ upon aging, suggests that organic aerosols are a critical component of PM in driving $\beta_{ext}$ in the Colorado

Front Range. However the correlation between dry $\beta_{ext}$ vs. OA was weakest in plumes with agricultural emissions (r = 0.17), suggesting OA had little impact on $\beta_{ext}$ in these plumes. The correlation coefficients for $\beta_{ext}$ vs. aerosol nitrate mass were strongest under the influence of O&G and agriculture and livestock emissions (r = 0.74 and 0.90 respectively), and weakest in the urban plumes (r = 0.37). These observations suggest that ammonia emissions that are co-located with O&G-related



activities in the Front Range play a significant role in controlling $\beta_{ext}$ in these air masses by enhancing the partitioning of nitric acid to the condensed phase. In fact, the average aerosol nitrate fraction over total nitrate (aerosol nitrate/ $HNO_3$ + aerosol nitrate) under agriculture and O&G plumes were $0.25 \pm 0.09$ and $0.11 \pm 0.10$, respectively, compared to $0.070 \pm 0.071$ in urban-influenced plumes. $\beta_{ext}$ was poorly correlated with sulfate aerosols in the region under the influence from all sources (r

$= 0.27, 0.33, 0.07, 0.08$ for urban, O&G, agriculture, and urban+O&G respectively), suggesting a low impact of sulfate aerosol and its precursors on dry $\beta_{ext}$ in the region.

Due to the higher hygroscopicity of inorganic salts compared to organics, contribution of sulfate and nitrate aerosols to the ambient $\beta_{ext}$ could be higher than what is discussed above. However, under the average ambient conditions encountered during FRAPPÉ (average RH~35-45%), the increase in ambient $\beta_{ext}$ due to aerosol hygroscopicity is not expected to be

significant (~20%) given the high organic fraction of 64-74% in urban-, O&G-, or urban + O&G-influenced plumes (Massoli et al. 2009). In agriculturally-influenced plumes, the influence of nitrate aerosol on ambient $\beta_{ext}$ will be greater because of the lower organic fraction and higher nitrate mass in these plumes, re-emphasizing the role of nitrate aerosol on $\beta_{ext}$ under such emissions.

### 3.3 Mass Extinction Efficiency

Mass extinction efficiency (MEE) is a function of the diameter of the particle, wavelength of attenuated light, and aerosol refractive index (Seinfeld et al. 2006). To further asses the impacts of aerosol sources on dry $\beta_{ext}$, MEE values, i.e., the ratio of the observed dry $\beta_{ext}$ to NR-PM$_1$ mass, in different air masses were estimated. For this analysis, MEE values were determined as the slope of the weighted linear ODR fits of $\beta_{ext}$ against NR-PM$_1$ mass, with weights representing standard deviations equal to the uncertainties in NR-PM$_1$ mass (30%) and $\beta_{ext}$ (10%). As indicated in Figure 5a-d, MEE values under the urban, O&G,

agriculture, and urban + O&G influence were ~$1.97 \pm 0.084, 1.88 \pm 0.064, 1.83 \pm 0.88$, and $3.30 \pm 0.094$ m$^2$ g$^{-1}$ with r values of $0.52, 0.78, 0.79$, and $0.59$, respectively. The overall MEE value in the Front Range, i.e., MEE observed for aerosol in all air mass types, was $2.66 \pm 0.024$ m$^2$ g$^{-1}$ (r= 0.74). As seen in Figure S2, different aerosol mass distributions were observed under different air mass types. For the mass distribution analysis, $d_{va}$ (vacuum aerodynamic diameter) was converted to $d_p$ (physical diameter) by dividing $d_{va}$ by the overall mass-weighted effective density ($\rho$), assuming $\rho$=1.25 g cm$^{-3}$ for OA, $\rho$ =1.75 g cm$^{-3}$

for ammonium sulfate and ammonium nitrate, and assuming that particles sampled by the mAMS were internally mixed (Jayne et al. 2000, Seinfeld et al. 2006). Scattering efficiency ($Q_{sp}$) which is a function of particle size and wavelength of light was estimated using Mie theory at $\lambda$=632 nm and a volume-weighted average refractive index of n=1.48, assuming n=1.45 for OA, n=1.53 for ammonium sulfate , and n=1.61 for ammonium nitrate (Seinfeld et al. 2006).

For typical urban air masses with data points along the line of the ODR fit of Figure 5a mass distributions were

dominated by organic aerosols in the size range of $d_p$=150-300 nm (Figure S3). This is consistent with previous observations for urban aerosol volume distributions with modes at the size range of $d_p$ ~ 200-500 nm (Seinfeld et al. 2006). To investigate the reason behind the low r value for the observations under the urban influence, data points below and above the ODR fit of Figure 5a were analyzed. Different mass distributions were observed for points below and above the ODR fits (Figure S2 a-





b). A mode of ~ 200-250 nm was seen for both sets of data while a second mode at ~ 400 nm was also observed for points above the ODR fit. Differences in the size distributions of urban aerosols might have contributed to the spread in the scatter plot and hence the lower r value in Figure 5a. Under O&G air masses (Figure S2c), individual mass distributions were broader, with modes for all species shifted to larger sizes ($d_p$ ~ 200-550 nm). In agriculturally-influenced air masses nitrate aerosols presented a significant mode in the size range of $d_p$ ~ 250-400 nm while OA species were concentrated on smaller sizes ($dp$ ~ 100-200 nm; Figure S2d).

As mentioned above, the average MEE value in the urban + O&G plumes was significantly higher than in the other air masses. Mass distributions with different MEE values, i.e., in plumes with data points below the ODR fit, MEE= 1.83 ± 0.80 m$^2$ g$^{-1}$, and above the ODR fit, MEE= 4.12 ± 0.69 m$^2$ g$^{-1}$ (Figure S4) were examined. Similar to the urban mass distributions discussed above, the mass distribution for the data points below the ODR fit (Figure S2e) was dominated by OA in the smaller size range (~ 90-110 nm), but it also included contributions from sulfates and nitrates in the larger size (~ 225-275 nm and ~ 430-550 nm). In contrast, the mass distribution in a plume with data points above the ODR fit (Figure S4f) had significantly higher contribution from OA in the size range of ~ 225-350 nm, showing a clear shift and OA growth to larger sizes with higher $Q_{SP}$. To gain a more comprehensive understanding of the impacts of variable size distributions on $\beta_{ext}$ and MEE values, PCASP number concentrations in the size range of 300-2000 nm ($N_{300-2000}$), where $Q_{SP}$ is more significant, were further examined. As shown in Figure 6 (also Figure S5), 60-90% of data obtained under the urban, O&G, and agriculture air masses, contained less than 15 particle/cm$^3$ in the size range of 300-2000 nm. This consistent observation of low number concentration in $N_{300-2000}$ supports the similarity of the low MEE values in these air masses. In contrast, 55-90% of data under the influence of urban+O&G mixed emissions, especially those with extinction values higher than the ODR fit, contained $N_{300-1000}$ more than 15 particle cm$^{-3}$. Higher contribution of larger particles in the urban+O&G mixed air masses is consistent with the higher MEE value observed in these air masses (Figure 5d). It is worth noting that the PCASP data were obtained at ambient conditions whereas the aerosol sampled by CAPS-PM$_{ex}$ and AMS were effectively dried upon sampling in the cabin. Therefore, the ambient size range of 300-2000 nm might translate to a slightly different dry size range due to day to day changes in the ambient relative humidity (RH) and variable aerosol hygroscopicity. However, this influence is expected to be minimal since average ambient RH levels were relatively low (~35-45%) and not changing drastically in different plumes (Table S1). Based on these examples, we conclude that the reason for the differing MEE values under different air mass types is primarily differences in the corresponding aerosol size distributions.

Next we examine the similarity of MEE values observed in the Colorado Front Range to previous measurements. MEE is the sum of the mass absorption and scattering efficiencies (MAE and MSE respectively), which both depend on particle size, refractive index, and wavelength of light (Seinfeld et al. 2006). Keeping in mind that in the presence of absorbing species, MEE is higher than mass scattering efficiency (MSE), in the absence of estimates of MEE in other regions, we present estimates of MSE from previous studies for comparison with the current MEE estimates in the Front Range. PM$_{2.5}$ scattering efficiencies at 550 nm in several ground based studies in urban commercial/ residential sites have typically been measured to be in the range of 2-3 m$^2$/g in (Chow et al. 2002, Hand et al. 2007). In such studies, the main aerosol sources contributing to the observed



PM$_1$ MSE were the automotive emissions and combustion processes. Although the contribution of elemental or black carbon to PM$_1$ mass during FRAPPE is unknown, similar to these previous studies, OA contributed the most to the NR-PM$_1$ mass in the Front Range and in comparison, the observed MEE values are consistent with the previous estimates of MSE.

### 3.4 Impacts of biomass burning (BB) emissions on optical extinction

During August 11 and 12, several wildfires were observed at Rocky Mountain National Park, near Tonahutu Creek Trail, 60 miles NW of Denver and Grand Mesa, Uncompahgre and Gunnison National Forest. BB gas-phase markers, namely hydrogen cyanide (HCN) and acetonitrile (CH$_3$CN) from TOGA and PTR-MS airborne data, respectively, were elevated in the boundary layer throughout the flights on Aug. 11-12 compared to non-biomass burning days (July 26, 29, 31 and August 2-3, 7-8, 15-16, 18). For example, during the BB days, HCN (CH$_3$CN) mean mixing ratio in the boundary layer was 516 $\pm$ 58 pptv (201 $\pm$ 44 pptv) whereas the boundary layer mean mixing ratio on non-BB days was 327 $\pm$ 59 pptv (148 $\pm$ 38 pptv). Since elevated levels of HCN and CH$_3$CN were not observed in individual plumes but rather throughout the boundary layer on Aug. 11-12, a regional influence of biomass burning emissions was suspected to be present in the Front Range during this time. Ground-based measurements of PM$_{2.5}$ from Denver-La Casa (39.78 N, -105.01W), Denver-CAMP (39.75 N, -104.99 W), and Denver-I25 (39.73 N, -105.02 W) sites were analyzed to assess the regional impact of wildfire emissions in the Front Range in terms of the contribution of PM$_{2.5}$ during the days of BB and non-BB. The time series of PM$_{2.5}$ mass concentrations at the sites described above, during days preceding and following the wildfires shows increases in mass concentration for PM$_{2.5}$ during the days of BB (Figure S6). The mean PM$_{2.5}$ mass concentrations during the times of 9 am to 7 pm local time at Denver, La Casa, Denver-CAMP, and Denver-I25 during non-biomass burning days were 5.61 $\pm$ 2.02, 6.01 $\pm$ 3.52, and 7.28$\pm$ 2.91µg m$^{-3}$, while mean mass concentrations increased to 9.47 $\pm$ 2.05, 11.51 $\pm$ 3.04, and 14.08 $\pm$ 4.68 µg m$^{-3}$, respectively, during the biomass burning days. As seen in Figure 7, the average daytime PM$_{2.5}$ mass concentration on BB days increased by 75-98% compared to the non-BB days confirming the regional influence of wildfires on the Front Range aerosol loadings. The contribution of organic aerosols to NR-PM$_1$ mass also increased from 80% during these events (Figure 3).

In addition to scattering of light by smoke particles, BB emissions of black carbon (BC) and brown carbon (BrC) can lead to significant absorption of the solar radiation in the visible and UV region; at 632 nm absorption by BrC is minimal (Lack et al. 2012, May et al. 2014). During Aug. 11-12, background values of airborne $\beta_{ext}$ were higher by 10-15 Mm$^{-1}$, suggesting the additional contribution to $\beta_{ext}$ from the wildfires. MEE and $\Delta\beta_{ext}/\Delta CO$ enhancement ratios were analyzed for days with and without the BB influence, using weighted linear ODR fit analysis, as explained previously. As seen in Figure 8, MEE and $\Delta\beta_{ext}/\Delta CO$ on Aug. 11-12 were ~ 30% and a factor of two greater, respectively, compared to days without the influence of BB. Although the AMS does not detect refractory materials such as BC due to the relatively low temperature of its vaporizer (600 °C), it is likely that on Aug. 11-12, BC emissions from the fires had resulted in elevated extinction values on a regional scale, resulting in higher MEE and $\Delta\beta_{ext}/\Delta CO$. The observed increase in MEE and $\Delta\beta_{ext}/\Delta CO$ on Aug. 11-12 suggest that regional BB emissions have at least a comparable impact on aerosol optical extinction and visibility in the Front Range relative to the local sources.



## 4 Conclusions

Airborne aerosol optical extinction (632 nm) and submicron non-refractory aerosol composition were measured during the summer in the Colorado Front Range to understand sources and processes that impact summertime visibility in the area. In assessing the role of atmospheric processing on $\beta_{ext}$, $\Delta\beta_{ext}/\Delta CO$ enhancement ratio increased under aged air masses by 50-60%.

The increase in $\Delta\beta_{ext}/\Delta CO$ in the aged air masses was accompanied by a factor of 3 increase in $\Delta OA/\Delta CO$, indicating that secondary formation of organic aerosols had significant impacts on the evolution of $\beta_{ext}$ in the Front Range. Correlation between $\beta_{ext}$ vs organic, nitrate, and sulfate aerosol mass under urban, O&G, agriculture, and urban + O&G mixed source influence were analyzed by linear regression fits. $\beta_{ext}$ best correlated with organic aerosols under the urban and O&G emissions and best correlated with nitrate aerosols under the O&G and agriculture influences. Correlation with sulfate was poor under

all air mass types. Estimated non-refractory mass extinction efficiency values for different air mass types ranged from 1.83-3.30 $m^2\ g^{-1}$, with the minimum and maximum values observed in agricultural and urban + O&G influenced air masses, respectively. Finally, aerosol components emitted from wildfires during the days of August 11 and 12 increased $\beta_{ext}$ background values by 10-15 $Mm^{-1}$ and resulted in higher MEE and $\Delta\beta_{ext}/\Delta CO$ values by about 30% and a factor of 2 respectively compared to non-biomass burning days, indicating that summertime visibility in the Front Range may significantly be impacted by

regional wildfires in addition to local sources.

### Acknowledgments

The authors thank Daniel Adams (UCR's CNAS Machine Shop) and technicians at the NCARs Research Aviation Facility for integration of the instruments on the aircraft rack, a smooth aircraft integration process, and support throughout the project,

Joshua Schwarz (NOAA-ESRL) for providing us the aircraft inlet system, Ron Cohen and Carly Ebben for providing the TD-LIF data, the Colorado Department of Public Health and Environment for funding and ground-based PM data, as well as the *USDA National Institute of Food and Agriculture, Hatch project Accession No. 233133* for additional funding support. Data used in this analysis may be obtained at http://www-air.larc.nasa.gov/cgi-bin/ArcView/discover-aq.co-2014?C130=1.

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





**Figures**

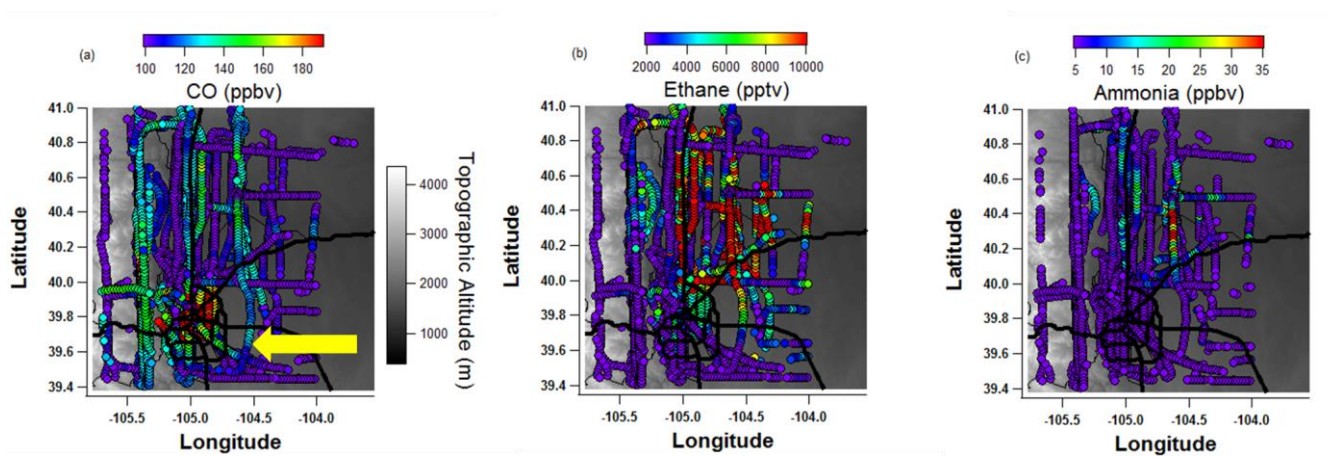

**Figure 1.** C-130 flight tracks in the Colorado Front Range, color coded with observed mixing ratios of (a) CO, (b) ethane, and (c) ammonia. The yellow arrow indicates the Denver metropolitan area. To the west of the Denver metropolitan area are the Rocky Mountain foothills depicted by the topographic color scheme.





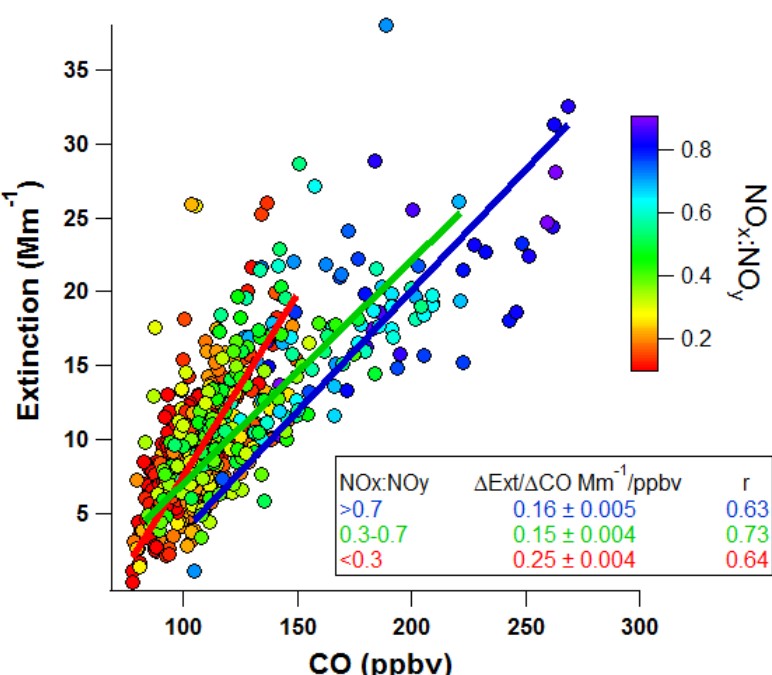

**Figure 2**. Orthogonal distance linear regression fits to extinction (Mm$^{-1}$) vs. CO (ppbv) under fresh (blue fit line), intermediately aged (green fit line), and aged air masses (red fit line).





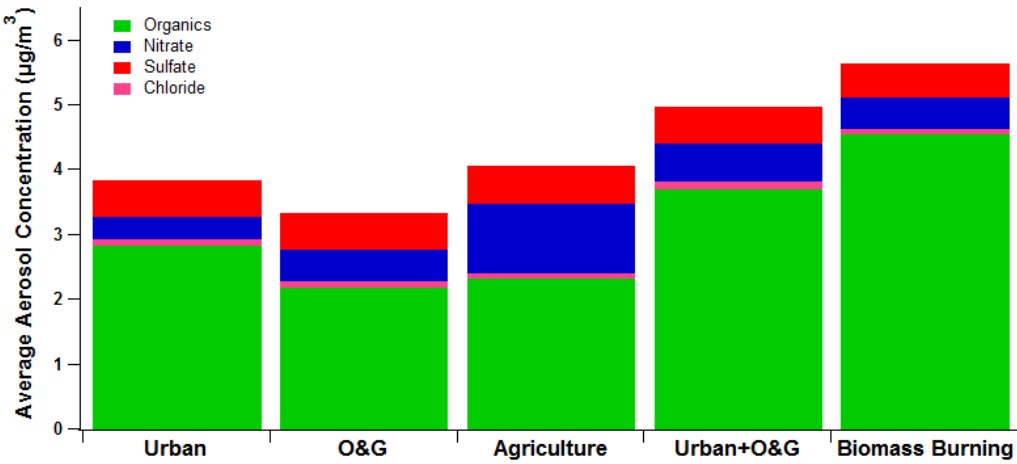

**Figure 3**. Average chemical composition (µg m$^{-3}$) of non-refractory aerosols under different air mass source.





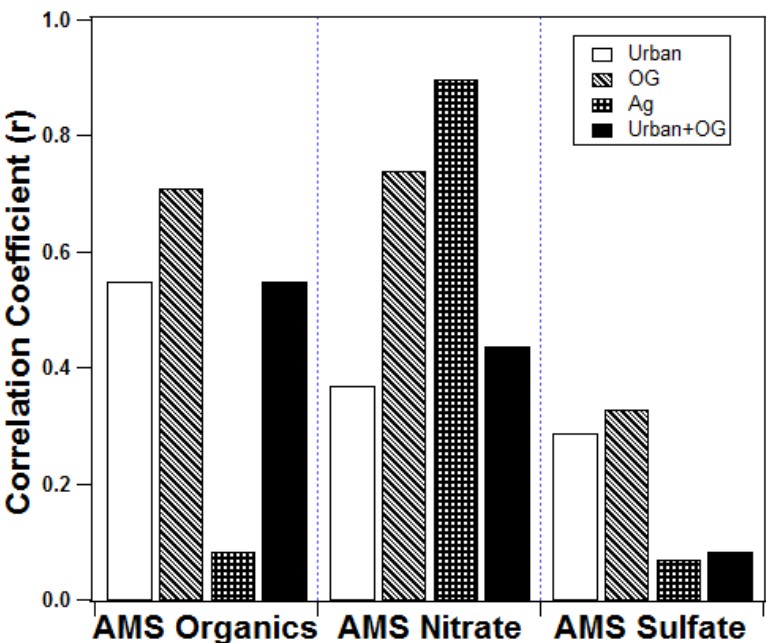

**Figure 4**. Correlations coefficients of $\beta_{ext}$ vs. aerosol composition under urban, O&G, agriculture, urban + O&G emissions.





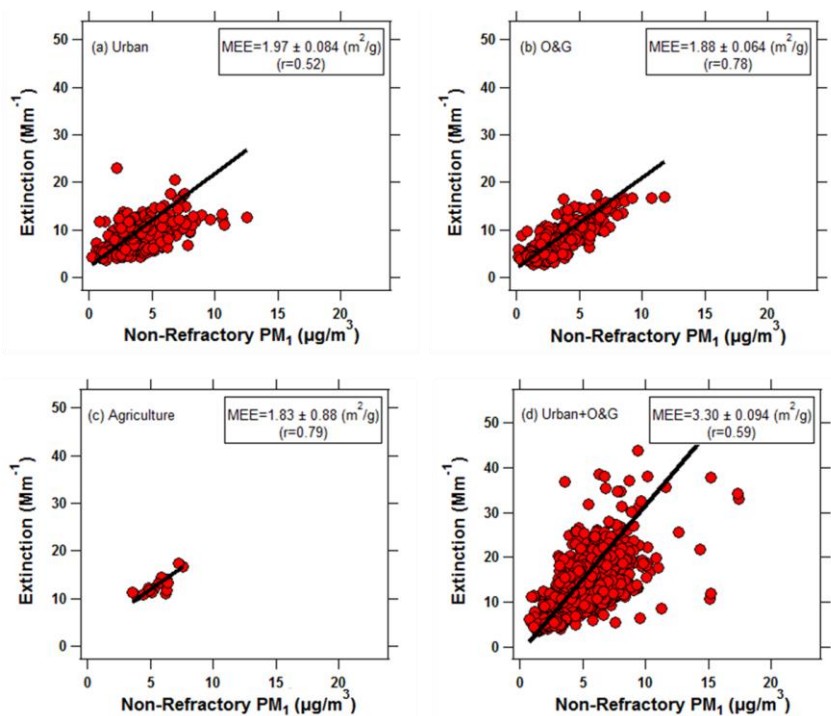

**Figure 5.** Mass extinction efficiencies (MEE) under (a) urban, (b) O&G, (c) agriculture, and (d) urban+O&G influence.




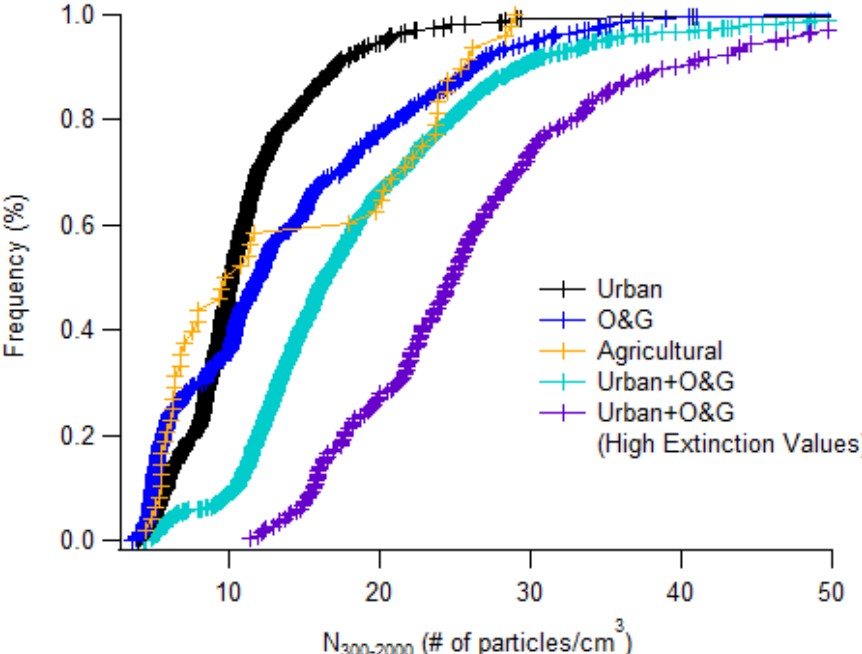

**Figure 6.** Cumulative particle number concentrations in the size range of 300-2000 nm in different air mass categories.





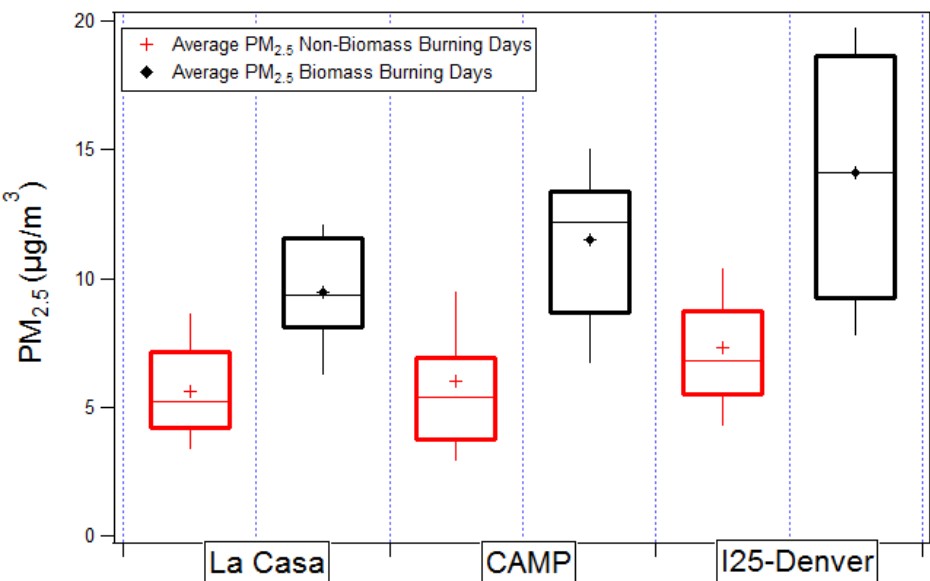

**Figure 7.** Daily (9 am - 7 pm local time) average PM$_{2.5}$ mass concentration for 3 monitoring sites for (a) non-biomass burning days of July 26, 29, 31 and August 02, 03, 07, 08, 15, 16, 18 and (b) biomass burning days of August 11 and 12. The whisker top, whisker bottom, box top and box bottom represents the 90$^{th}$, 10$^{th}$, 75$^{th}$, and 25$^{th}$ percentiles.





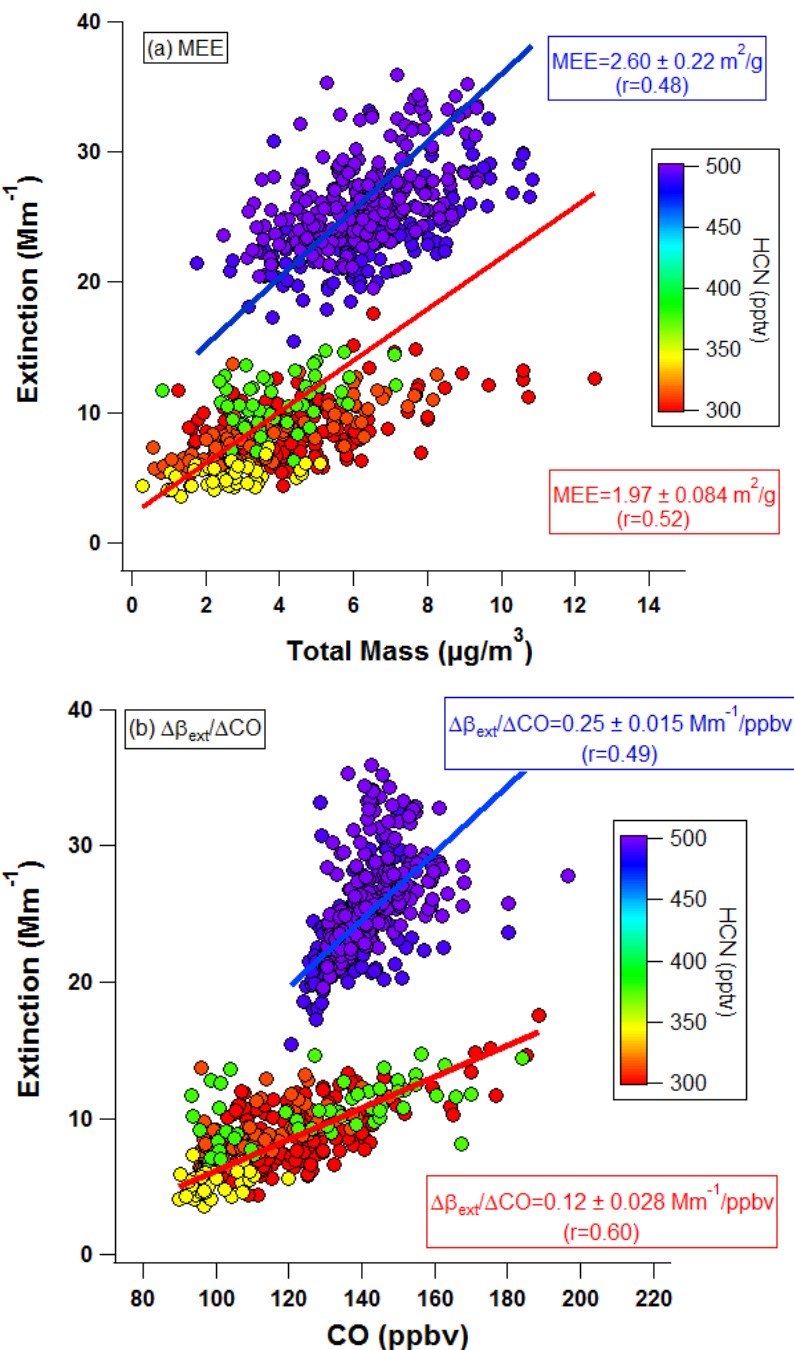

**Figure 8.** Orthogonal distance linear regression fits to (a) extinction (Mm$^{-1}$) vs. total NR-PM$_1$ mass (µg/m$^3$) and (b) extinction (Mm$^{-1}$) vs. CO (ppbv). Data points are color coded with the average HCN mixing ratio for non-biomass burning and biomass burning days.