# Peer review of "Aerosol Optical Extinction during the Front Range Air Pollution and Photochemistry Éxperiment (FRAPPÉ) 2014 Summertime Field Campaign, Colorado U.S.A."

_Atmospheric Chemistry and Physics, 2016_

## Referee Comment (RC1) · Anonymous Referee #1 · 10 May 2016

This paper discusses observed light extinction of nominally PM2 particles measured over the Colorado front range during the FRAPPE aircraft study. The authors assert that this paper provides an updated assessment on the Denver Brown Cloud. This is a worthwhile topic and the paper is suitable for publication in ACP. However, there are number logical inconsistencies, important missing information, and other issues that must be addressed, including:

-particle size range and RH of the extinction measurement is not well characterized making the data of questionable value (ie, how to compare to other studies and how to

apply to ambient conditions).

-mismatch between AMS and extinction measured particle size ranges.

-the justification for the use of extinction versus CO to compare extinction versus photochemical age for all combined sources.

Also, given the discussion in the Introduction that the motivation of this work was to take a new look at the Denver Brown cloud, it is rather odd that this is never done. It would be insightful to add a section on comparing/contrasting these results to earlier studies; has visibility improved, have sources that contribute to visibility reduction changed, etc.

Specific comments:

Why is there no discussion of any anthropogenic gases that may contribute to the Denver Brown Cloud, either in past studies or this study? Are they not important (give numbers to support). Are they included in the reported extinction measurement, or subtracted out with the blank correction?

Page 3 and throughout; specifically note that the altitudes give are above sea level (I assume), not surface?

Page 4, line 18; the CAPs(ext) did not have a size selective inlet; apparently upper size limit is controlled by only inlet/sample line transmission efficiencies? Discuss in more detail, specifically how well is the size range of particles contributing to the measured extinction really known (give the uncertainty, my suspicion is that it is large of it is bases solely on calculated inlet and sample line transmissions). What are the implications of this uncertainty (the size distribution was measured so a quantitative estimate should be possible).

How does one handle the mismatch in particle sizes sampled with the AMS and CAPs? This could have impacts on much of the reported data, depending on the shape of the size distribution. Add a discussion.

No discussion on RH (or T) of sample in the CAPS? RH variability could have a large effect on extinction. In the paper it is referred to as dry extinction, but RH is never given? It appears that the authors are just assuming the particles are dry since the ambient RH was low and the particles heated in the inlet/sample line. Much more detail, along with possible differences in LWC of sampled and ambient aerosol, should be considered. Note, at the least one could estimate the RH in the CAPS assuming the aerosol has reached cabin T, if one knows the ambient RH and T. Claiming a dry extinction measurement really requires reporting actual RH in the CAPs.

Page 4 line 22, typo, intends or just tends?

Re. Fig 2 and the general idea of looking at extinction vs CO: The logic behind the graph and more details may be needed. First, is this data just for well defined plumes or include all data, except biomass burning (ie, it includes urban and agri, urban+O&G, and O&G)? Second, this plot is predicted on a correlation between extinction and CO; that is that the components driving extinction and CO are co-emitted in all sources included in this plot. This appears to be the case, but it is curious why this is so if it includes all these various sources. That is, if this plot is for all sources combined, why do they all have similar Ext/CO ratios (ie, only a function of age)? Maybe this plot is mainly driven by urban emissions. This would also mean that most of the aging is just due to OA aging. Fig 3 would support this, in a general sense. Why not use a PMF analysis and look at evolution of specific AMS OA factors?

Why lump all the data together in this plot since it is more valid for a plume from a specific source; wouldn't graphs like this for each specific source make more sense, or maybe just focus on the urban data? Also, one would expect that some components that contribute to extinction, such as sulfate and nitrate would not be correlated with CO and so not appropriate to include sources with high emissions of these components in this analysis. Maybe this accounts for much of the scatter? One might also give the overall r2 between extinction and CO (ie not segregated by age) in Fig 2, and finally, why the different intercepts in Fig 2?

Fig 3, any estimates on potential bias in the composition data due to sampling only submicron non-refractory aerosol with the AMS? In some sources this could lead to substantial bias, eg, the AMS would not measure more refractory nitrate salts that could be present in some of the sources (eg, NaNO3, Ca(NO3)2, . . .).

Why is there so much OA associated with agri emissions?

Page 6 last line, the assumption is being made that nitrate formation is controlled by NH3 concentrations through partitioning of nitric acid. What is the justification for this? The process is actually likely to be much more complicated as it depends on the pH of the aerosol, which in turn depends on the amount of mineral dust and sulfate also present; it doesn't just depend on NH3 concentration. Also, given that NH3 was measured, one could be more specific and quantify the differences in NH3 levels in the various source regions.

Fig 6, how can there be so few particles (generally less than 40 or so particles per cm3 of air, get mass concentrations are up to 15 to 20 ug/m3? Seems very odd.

Fig 8, the correlations are not that good, total mass explains only 25 to 35% of the extinction variability (r2), so are the regressions really meaningful (comparisons of slopes for each plot)?

———————————————————

---

## Referee Comment (RC2) · Anonymous Referee #2 · 28 Jun 2016

Review of Aerosol Optical Extinction during the Front Range Air Pollution and Photochemistry Éperiment (FRAPPÉ) 2014 Summertime Field Campaign, Colorado U.S.A.

Overall Comments This paper describes relationships between mass measurements from the mAMS and extinction measurements from a CAPS-PMex instrument in the Front Range of Colorado. The authors derive Mass Extinction Efficiencies for a number of different airmass types (urban, oil and gas, agricultural) based on gas-phase tracer measurements from the study. The paper presents some interesting, relevant results and is appropriate for publication in ACP after some significant revisions described

below. The major problems with the submitted manuscript are

1.) The error estimates for the CAPS-PMex are not explained or justified for this study, only a reference is given. The error estimate of 10% seems much too small for small extinction measurements (<10 Mm-1) given that the authors observed significant baseline shifts. 2.) The authors see poor correlation between mass measured by the AMS and extinction. This poor correlation is stated to be due to changes in size distribution. However, in some of the plume types, especially urban, there does not appear to be a dramatic difference in the size-distribution of low and high MEE plumes. If the authors are going to conclude that size distribution is responsible for widely varying MEE, this needs to be backed up with some Mie modeling that shows the observed shifts in size distribution can cause the observed shifts in MEE. It is also possible that varying size cutpoints between the mAMS and CAPS-PMex are significant an this needs to be discussed.

Additional comments are given below.

Comments on Introduction 1.) It is stated that, "it has been observed that the important contributors to light scattering in the Colorado Rocky Mountains are particulate matter from the urban emissions" Certainly there are dust and smoke events that cause visibility degradation too, in fact the authors show a smoke event. This statement is far too broad.

2.) The authors discuss the wintertime phenomenon of the Denver Brown Cloud but do not discuss summertime visibility issues. Some background on typical extinction, or at least PM2.5 mass in the summertime in Denver is needed.

3.) The statement "twofold increase in natural resource extraction" is vague. Is this an increase in wells, in gas production, in oil production? This is relevant because each has a different implication for air quality.

4.) The authors state that emissions from oil and gas include "emissions from industrial

equipment known to emit...particulate matter." Are they referring to BC from diesel engines? There needs to be clarification here because there are not typically large primary emissions of particulate matter from oil and gas operations.

5.) I don't see a basis for the authors to conclude that photochemical production of ozone is significantly impacted by oil and gas operations. There's a big jump from OH reactivity in the winter to ozone production in the summer and no evidence to back up this logical leap. This also has little to do with the paper, I would suggest removing the last line of page 2 and the first line of page 3.

Comments on Instrumentation and Methodology 1.) The description of the CAPS-PMex operating principles is fairly poor. It makes it seem that the extinction is detected by monitoring the phase shift that occurs during 1 transit of the light from the first mirror to the second. This is far from accurate, the entire point of the instrument is to create a very long effective pathlength.

2.) Line 4 page 4: An averaging time needs to be given for uncertainty and detection limit statements.

3.) Why were baseline values only interpolated when the shift was more than 5 Mm-1? I see no reason not to always interpolate them. It also needs to be explained why the baseline would shift that much. Given that most reported measurements are <20 Mm-1, this would seem to be a big issue.

4.) Error analysis for the CAPS seems to need significantly more attention in general. 10% is clearly not a correct estimate of error in the extinction coefficient when the average extinction is something around 15 inverse Mm-1 and you have a baseline shift of 5 Mm-1. The authors need to do a careful assessment of error in the extinction measurement.

5.) Why is particulate nitrate not considered in the NOx/NOy ratio? This seems odd given the authors are measuring particulate nitrate with the mAMS.

[Figure]

6.) A PCASP does not seem to be the best instrument for submicron aerosol analysis. Can the authors comment on why an SMPS or UHSAS was not used in addition to the PCASP? If these instruments were not on the C-130, this needs to be stated.

7.) There needs to be a discussion of the aircraft inlets and size cutpoints for the various instruments. This is especially critical for the PCASP vs. CAPS-PMex vs. mAMS given that the instruments have different cut points around a micron.

8.) There needs to be a discussion of how the mAMS data was corrected for collection efficiency, especially given that agricultural plumes had significant nitrate.

Comments On Results and Discussion 1.) Figure S1 appears to show roads but this is not described in the caption

2.) Again, error analysis is a problem. The uncertainty in Bext being stated to be 10% when the authors state that the baseline shifts by over 5 Mm-1 and that the 1-sigma detection limit is 1 Mm-1 seems completely inconsistent. Values of Bext of less than 5 Mm-1 are shown and the authors are suggesting the error is < 0.5 Mm-1 for these measurements but this is half of the 1-sigma detection limit and does not account for baseline shifts. The authors make a major point that they are doing ODR because they want to weight according to the errors in the instruments, but the errors seem incorrect.

3.) Page 5 Line 17. The delta ext/co ratios have error bars associated with them, but the authors don't explain what they are. Are these 1-sigma intervals on the slope of the fitted line? For individual measurements the errors would be much larger than these (∼11%), I am surprised at how small they are given the poor r of the fits. Also, the authors remove the error in the intercept from the discussion even though this is probably significant.

4.) The authors need to explain how they arrived at the criteria of 0.7, 0.3-0.7, and <0.3 for NOx:NOy ratios.

5.) Page 5 Line 29. The authors state that the "bulk" of the organic aerosol mass is

[Figure]

SOA because the enhancement ratio of organic aerosol increases significantly for aged aerosol. This is not adequate reasoning. The aerosol could be unrelated to CO (from non-combustion sources), as the authors themselves point out. The increasing Bext could be the result of increasing ammonium nitrate when plumes move farther from the city. The authors need to explain what fraction of aerosol mass is accounted for by the OA:CO enhancement or something in addition to the current argument if they are going to make this sweeping statement. Also, the word "bulk" needs to be quantified in some way.

6.) Page 6 Line 16. Correlation coefficients of .55 are not "strong" 7.) Page 6 Line 23. Nitric acid + aerosol nitrate is not "total" nitrate, this ignores organic nitrates. 8.) The relatively weak correlations in urban plumes between Bext and any of the aerosol species (all have r less than 0.55) mass concentrations suggests that there was poor correlation between total AMS mass and Bext. In fact, this is born out in figure 4. This is very confusing and needs to be addressed, especially given the very good correlation between Bext and nitrate in agricultural plumes. The authors suggest it may be caused by a change in size distribution (Figure S2 a-b), but it appears to me that the total mass distributions in these two figures are fairly similar, even though the organic distributions are a little different. Are the authors missing an important aerosol species? Alternaltively, is the mismatch in measured aerosol size between the CAPS and AMS significant (Figure S2 suggests this may be the case and that a careful application of the lens transmission is required)? If the authors are going to rest on the size-distribution argument then they need to implement a Mie model of the two distributions shown in S2 and demonstrate that they yield dramatically different Bext for the same total aerosol mass. The size distribution argument for urban + OG is more convincing.

9.) In Figure 4 the removal of the intercept and its associated error is again not mentioned, even though it is a significant issue given the lines don't go through 0,0 and the r values are fairly low. I am again of the opinion that the stated error in the slopes does

not accurately represent the error in this slope.

10.) Page 8 Line 7-8. If the authors are going to make this broad conclusion they need to some basic Mie modeling to convince the reader that these shifts in size distribution can generate MEE from 1.8 to 4.1, this does not seem obvious.

11.) In Figure 7a "total mass" needs to be changed to "total non-refractory PM-1 mass", though even the PM-1 designation is questionable with AMS data.

12.) In Figure 7a the fit to the non-biomass burning emissions does not seem to track the data. The authors need to discuss this. Perhaps it is due to including moderate HCN data points.

13.) Again in Figure 7 the intercepts are removed and not discussed, they need to be discussed because they are very different between biomass burning and non-biomass burning and this has physical meaning.

---

## Author Comment (AC1) · 8 Aug 2016

We appreciate having received detailed comments from the reviewers. We have revised the manuscript accordingly. Below, you will find our response and the summary of our approach, highlighted in *red,* with modifications to the manuscript highlighted *in bold*:

*Referee #1:*

This paper discusses observed light extinction of nominally PM2 particles measured over the Colorado Front Range during the FRAPPE aircraft study. The authors assert that this paper provides an updated assessment on the Denver Brown Cloud. This is a worthwhile topic and the paper is suitable for publication in ACP. However, there are number logical inconsistencies, important missing information, and other issues that must be addressed, including:

-particle size range and RH of the extinction measurement is not well characterized making the data of questionable value (ie, how to compare to other studies and how to apply to ambient conditions).

*The reviewer has raised a good point. We have estimated the RH in the CAPS-PM$_{ex}$ unit, using the measured ambient temperature and RH assuming aerosols had equilibrated to the temperature within the instrument. Our results indicate that on average the RH in the CAPS was 20 ± 7% with a range of 15-30% while ambient RH was on average 44±17%.*
*We have addressed this issue in section 2.2, paragraph 4 by adding the following sentences:* **"Based on the ambient RH and temperature and the temperature within the CAPS-PM$_{ex}$ extinction cell, and assuming that aerosols had equilibrated to the conditions within the measurement cell, the CAPS-PM$_{ex}$ measurements for the flights discussed here represent extinction values at an average RH of 20 ± 7 % (range of 15-30%)."**

-mismatch between AMS and extinction measured particle size ranges.

*In order to determine whether the discrepancy between the aerosol size ranges being sampled by the CAPS-PM$_{ex}$ and AMS had a significant impact on our analysis or not, we have used the size distributions from the Passive Cavity Aerosol Spectrometer Probe (PCASP) instrument on board the C-130 to estimate the ambient scattering coefficients. By using a nominal refractive index of 1.5, estimated scattering (i.e., extinction, while assuming purely scattering aerosols) coefficients were calculated using the measured size distributions up to 800 nm (upper "true" size cut of the AMS) and 2000 nm (upper size cut of the inlet, and thus CAPS-PM$_{ex}$). The slopes of the scatter plots of the estimated scattering coefficients for PM$_{0.8}$ vs. PM$_2$ under the influence of urban, O&G, agricultural, and urban+O&G slopes were 0.95 ± 0.01, 1.0 ± 0.002, and 1.0 ± 0.01, 0.92± 0.01, respectively, indicating that the majority of the signal contribution to extinction originated from aerosols in the size range of the mAMS. We also note that the slope values mentioned above were not highly sensitive to the choice of the refractive index. Changing the refractive index from 1.48 to 1.52 changed the slope values by at most 4%.*
*The following has been added to the text in section 2.2, paragraph 3 to address this issue:* **"Ambient aerosol size distributions were measured on-board the C-130 by a Passive Cavity Aerosol Spectrometer Probe (PCASP). Estimated extinction values using Mie calculations with a nominal refractive index of 1.5 and the measured PCASP size distributions indicated that particles smaller than 800 nm captured >92% of PM$_2$ extinction values, confirming that the majority of the extinction signal originated from aerosols in the size range of the mAMS. We note that the calculated extinction coefficients were not highly sensitive to the choice of refractive index; only a 4% decrease in the slope of scattering coefficients from PM$_{0.8}$ vs. PM$_2$ was observed by increasing the refractive index from 1.48 to 1.52."**

-the justification for the use of extinction versus CO to compare extinction versus photochemical age for all combined sources.

*We agree with the reviewer that it makes more sense to limit the data in Figure 2 to plume types where aerosol precursors are co-emitted with CO. Therefore, Figure 2 has now been updated to include data from urban emissions only, with the modified definition provided in Section 3.1 as* **"…plumes with enhancement of CO over the background (105 ppbv, as defined by the mode in the frequency distribution of CO in the Front Range boundary layer) while $\Delta C_2H_6/\Delta CO < 20$ pptv ppbv$^{-1}$).** *We also had to remove data from July flights due to lack of optimum quantitative quality of CO data during those flights, that was reflected on the data archive site after initial submission of the paper. With these changes, aging categories needed to be updated to $NO_x:NO_y > 0.5$ and $<0.5$ to represent relatively fresh and aged plumes, respectively, in order to include enough data points in each category. Despite these changes, the conclusions remain the same that with the reduction in $NO_x/NO_y$ and increase in photochemical aging, the enhancement ratio of $\Delta\beta_{ext}/\Delta CO$ increased significantly (by ~54%).*

Also, given the discussion in the Introduction that the motivation of this work was to take a new look at the Denver Brown cloud, it is rather odd that this is never done. It would be insightful to add a section on comparing/contrasting these results to earlier studies; has visibility improved, have sources that contribute to visibility reduction changed, etc.

*Since summertime extinction data from previous field studies in the Colorado Front Range are not available, we have used transmissometer extinction data, provided by the Colorado Department of Public Health, to consider monthly average values of extinction measured in Downtown Denver for the months of July and August during 2001-2014.*

*We have included the following sentences in section 3.3, paragraph 4 describing the observations:* **"In response to the wintertime haze episodes observed in the region, the State of Colorado has implemented a visibility standard based on total optical extinction of 76 Mm$^{-1}$ at 550 nm, averaged during a 4-hr period when ambient RH is less than 70% (Ely et al. 1993). Total optical extinction measurements are provided by the Colorado Department of Public Health and Environment's transmissometer, installed in Downtown Denver. We have assessed the average monthly total extinction coefficients for August of 2001-2014 to examine the recent trend in summertime extinction and visibility in the region. Averaged monthly values varied from 40 to 80 Mm$^{-1}$, and no significant trend was observed since 2001."**

Specific comments:

Why is there no discussion of any anthropogenic gases that may contribute to the Denver Brown Cloud, either in past studies or this study? Are they not important (give numbers to support). Are they included in the reported extinction measurement, or subtracted out with the blank correction?

*Based on wintertime optical extinction measurements in 1978, (Groblicki et al. 1981)estimated that gaseous scattering and $NO_2$ absorption each contributed to 7% of total extinction at 550 nm. As described in Section 2.2, CAPS-PM$_{ex}$ provides only measurements of aerosol optical extinction since frequent filtered-air samples are collected during normal operation to subtract the background gaseous contributions to extinction. Following Groblicki's derivation of absorption coefficient at 550 nm using $NO_2$ mixing ratios and since $NO_2$ absorption cross section at 632 nm is about 10× lower than at 550 nm (Schneider et al, 1987), estimated average $NO_2$ absorption at 632 nm in the Front Range was less than 0.1 Mm$^{-1}$. Therefore, although the reported measurements of extinction are for aerosol particles, contribution of anthropogenic gases to total extinction at 632 nm in the Front Range is negligible.*

*The following has been added to Section 2.2:* **"It is worth reiterating that anthropogenic gases such as nitrogen dioxide have minimal effect on the measured $\beta_{ext}$ at 632nm since regular baseline**

*corrections based on sampled filtered air were applied to the data. Given the average mixing ratio of NO₂, the parameterization by Groblicki et al. (1981) for estimating NO₂ absorption at 550 nm, and the factor of 10 smaller value of NO₂ absorption cross section at 632 nm compared to 550 nm (Schneider et al, 1987), we estimated the average absorption of NO₂ to be ~0.1 Mm⁻¹, indicating a minor contribution of NO₂ to total extinction at 632 nm."*

Page 3 and throughout; specifically note that the altitudes give are above sea level (I assume), not surface?

> *In section 2.1, we have now noted that altitude is above sea level.*

Page 4, line 18; the CAPs(ext) did not have a size selective inlet; apparently upper size limit is controlled by only inlet/sample line transmission efficiencies? Discuss in more detail, specifically how well is the size range of particles contributing to the measured extinction really known (give the uncertainty, my suspicion is that it is large of it is bases solely on calculated inlet and sample line transmissions). What are the implications of this uncertainty (the size distribution was measured so a quantitative estimate should be possible). How does one handle the mismatch in particle sizes sampled with the AMS and CAPs? This could have impacts on much of the reported data, depending on the shape of the size distribution. Add a discussion.

> *Since the data presented in the manuscript were limited to the boundary layer, variations in the transmission efficiency of the inlet were really minor. We have calculated transmission efficiency of the inlet given a range of ambient pressures (760-860 mbar) and ambient temperatures (15-30 °C) representative of the BL; the 50% size cut for these conditions was 2.05± 0.05 um. As further discussed below, most of the signal contribution to aerosol extinction was from much smaller particles (<800 nm), so minor variations in the transmission of ~2 um particles could not pose significant uncertainties in the measurements.*
>
> *In order to determine whether the discrepancy between the aerosol size ranges being sampled by the CAPS-PM$_{ex}$ and AMS had a significant impact on our analysis or not, we have used the size distributions from the Passive Cavity Aerosol Spectrometer Probe (PCASP) instrument on board the C-130 to estimate the ambient scattering coefficients. By using a nominal refractive index of 1.5, estimated scattering (i.e., extinction, while assuming purely scattering aerosols) coefficients were calculated using the measured size distributions up to 800 nm (upper "true" size cut of the AMS) and 2000 nm (upper size cut of the inlet, and thus CAPS-PM$_{ex}$). The slopes of the scatter plots of the estimated scattering coefficients for PM$_{0.8}$ vs. PM$_2$ under the influence of urban, O&G, agricultural, and urban+O&G slopes were 0.95 ± 0.01, 1.0 ± 0.002, and 1.0 ± 0.01, 0.92± 0.01, respectively, indicating that the majority of the signal contribution to extinction originated from aerosols in the size range of the mAMS. We also note that the slope values mentioned above were not highly sensitive to the choice of the refractive index. Changing the refractive index from 1.48 to 1.52 changed the slope values by at most 4%.*
>
> *The following has been added to the text in section 2.2, paragraph 3 to address this issue:*
> **"Ambient aerosol size distributions were measured on-board the C-130 by a Passive Cavity Aerosol Spectrometer Probe (PCASP). Estimated extinction values using Mie calculations with a nominal refractive index of 1.5 and the measured PCASP size distributions indicated that particles smaller than 800 nm captured >92% of PM$_2$ extinction values, confirming that the majority of the extinction signal originated from aerosols in the size range of the mAMS. We note that the calculated extinction coefficients were not highly sensitive to the choice of refractive index; only a 4% decrease in the slope of scattering coefficients from PM$_{0.8}$ vs. PM$_2$ was observed by increasing the refractive index from 1.48 to 1.52."**

No discussion on RH (or T) of sample in the CAPS? RH variability could have a large effect on extinction. In the paper it is referred to as dry extinction, but RH is never given? It appears that the authors are just assuming the particles are dry since the ambient RH was low and the particles heated in

the inlet/sample line. Much more detail, along with possible differences in LWC of sampled and ambient aerosol, should be considered. Note, at the least one could estimate the RH in the CAPS assuming the aerosol has reached cabin T, if one knows the ambient RH and T. Claiming a dry extinction measurement really requires reporting actual RH in the CAPs.

*The reviewer has raised a good point. We have estimated the RH in the CAPS-PM$_{ex}$ unit, using the measured ambient temperature and RH assuming aerosols had equilibrated to the temperature within the instrument. Our results indicate that on average the RH in the CAPS was 20 ± 7% with range of 15-30% while ambient RH was on average 44±17%.*

*We have addressed the issue on RH in section 2.2 paragraph 4 by adding the following sentences:* **"Based on the ambient RH and temperature and the temperature within the CAPS-PM$_{ex}$ extinction cell, and assuming that aerosols had equilibrated to the conditions within the measurement cell, the CAPS-PM$_{ex}$ data discussed here represent extinction values at an average RH of 20 ± 7 % (range of 15-30%)".**

Page 4 line 22, typo, intends or just tends?

*The sentence has been rephrased.*

Re. Fig 2 and the general idea of looking at extinction vs CO: The logic behind the graph and more details may be needed. First, is this data just for well defined plumes or include all data, except biomass burning (ie, it includes urban and agri, urban+O&G, and O&G)? Second, this plot is predicted on a correlation between extinction and CO; that is that the components driving extinction and CO are co-emitted in all sources included in this plot. This appears to be the case, but it is curious why this is so if it includes all these various sources. That is, if this plot is for all sources combined, why do they all have similar Ext/CO ratios (ie, only a function of age)? Maybe this plot is mainly driven by urban emissions. This would also mean that most of the aging is just due to OA aging. Fig 3 would support this, in a general sense. Why not use a PMF analysis and look at evolution of specific AMS OA factors? Why lump all the data together in this plot since it is more valid for a plume from a specific source; wouldn't graphs like this for each specific source make more sense, or maybe just focus on the urban data? Also, one would expect that some components that contribute to extinction, such as sulfate and nitrate would not be correlated with CO and so not appropriate to include sources with high emissions of these components in this analysis. Maybe this accounts for much of the scatter? One might also give the overall r2 between extinction and CO (ie not segregated by age) in Fig 2, and finally, why the different intercepts in Fig 2?

*We agree with the reviewer that it makes more sense to limit the data in Figure 2 to plume types where aerosol precursors are co-emitted with CO. Therefore, Figure 2 has now been updated to include data from urban emissions only, with the modified definition provided in Section 3.1 as* **"…plumes with enhancement of CO over the background (105 ppbv, as defined by the mode in the frequency distribution of CO in the Front Range boundary layer) while ΔC$_2$H$_6$/ΔCO < 20 pptv ppbv$^{-1}$).** *We also had to remove data from July flights due to lack of optimum quantitative quality of CO data during those flights, that was reflected on the data archive site after initial submission of the paper. With these changes, aging categories needed to be updated to NO$_x$:NO$_y$ > 0.5 and <0.5 to represent relatively fresh and aged plumes, respectively, in order to include enough data points in each category.*

*Carrying out PMF analysis is outside the scope of this paper.*

*Indeed the correlation coefficients improved from r ~0.6-0.7 to r ~0.85-0.9, when excluding the non-urban plumes from this plot, confirming that some species that contributed to β$_{ext}$ were not co-emitted with CO.*

*The different intercepts observed when considering all plume types would have suggested different background levels of β$_{ext}$ due to inclusion of all aerosol source types in the plot. With the current*

*modification of including data from only the urban plumes, the fresh and aged fitted lines cross similar $\beta_{ext}$ values (6.0-6.7 Mm$^{-1}$) at the background CO level of 105 ppbv.*

Fig 3, any estimates on potential bias in the composition data due to sampling only submicron non-refractory aerosol with the AMS? In some sources this could lead to substantial bias, eg, the AMS would not measure more refractory nitrate salts that could be present in some of the sources (eg, NaNO3, Ca(NO3)2, : : :).

*On average, less than 0.5 µg/m$^3$ of Ca$^{2+}$ plus Mg$^{2+}$ (Na$^+$ concentrations were not reported) was present in the PM$_1$ aerosols as measured by a PILS aboard the C130; therefore, contribution of refractory salts is not expected to be significant.*

Why is there so much OA associated with agri emissions?

*The organics that were associated with aerosols observed in agricultural plumes were not originating from agriculture emissions since no significant enhancement in OA was observed while crossing such plumes. Therefore, the organics merely represent the composition of the background aerosol onto which agricultural emissions were superimposed.*

Page 6 last line, the assumption is being made that nitrate formation is controlled by NH3 concentrations through partitioning of nitric acid. What is the justification for this?
The process is actually likely to be much more complicated as it depends on the pH of the aerosol, which in turn depends on the amount of mineral dust and sulfate also present; it doesn't just depend on NH3 concentration. Also, given that NH3 was measured, one could be more specific and quantify the differences in NH3 levels in the various source regions.

*It is true that formation of ammonium nitrate depends on aerosol pH and other components of aerosol. As mentioned above, dust components of aerosol based on PILS data were minor. Additionally, AMS composition indicates that chloride and sulfate levels were very uniform in different air masses. The most variable parameter that could have an impact on aerosol composition was NH$_3$ levels, with average values of 1.41 ± 1.2 ppbv, 2.75 ± 1.88 ppbv, 8.21 ± 2.06 ppbv, and 5.47 ± 1.81 ppbv in urban, O&G, agriculture, and urban+O&G plumes, respectively. The following has been added in Section 3.2 to support our hypothesis: "**Aerosol nitrate formation depends on ambient conditions (temperature and relative humidity), relative mixing ratios of nitric acid and ammonia, as well as aerosol composition and pH (Seinfeld and Pandis 2006, Weber et al. 2016). With uniform concentrations of sulfate aerosol and small contribution of chloride and dust components to the Front Range fine aerosol mass, variability in aerosol pH was not expected to be high. Furthermore, there was no specific trend in temperature or relative humidity in different plume types. On the other hand, mixing ratios of ammonia were observed to be variable in the different air masses, with average values of 1.41 ± 1.2 ppbv, 2.75 ± 1.88 ppbv, 8.21 ± 2.06 ppbv, and 5.47 ± 1.81 ppbv in urban, O&G, agriculture, and urban+O&G plumes, respectively.**"*

Fig 6, how can there be so few particles (generally less than 40 or so particles per cm3 of air, get mass concentrations are up to 15 to 20 ug/m3? Seems very odd.

*The reviewer might have misread the horizontal axis in Fig 6. The axis represents the number of aerosol particles in 300-2000 nm size range and not the total number of fine aerosols. The low number of aerosols in the larger size bins just indicates that the majority of ambient particles were at sizes smaller than 300 nm, which is typically the case. This figure is no longer included in the manuscript.*

Fig 8, the correlations are not that good, total mass explains only 25 to 35% of the extinction variability (r2), so are the regressions really meaningful (comparisons of slopes for each plot)?

*We have updated this Figure to include data with masks designating the 4 plume types that have been examined in detail in the paper. This has greatly improved the correlations of $b_{ext}$ and NR-PM$_1$ mass, with r values ~0.75, as well as the trends of the weighted ODR fits. Please note that because BB also contributes to atmospheric CO, we decided the conclusions drawn from the scatter plot of $\beta_{ext}$ vs. CO in the presence and absence of BB could not be as robust as desired and have therefore deleted panel b.*

*The following text in Section 3.4 has been updated accordingly:* ***"MEE values were analyzed for days with and without the BB influence, using weighted linear ODR fit analysis, as explained previously. As seen in Figure 8, average MEE on Aug. 11-12 was ~70% greater compared to days without the influence of BB (3.65±1.16 m$^2$/g vs. 2.24±0.71 m$^2$/g). Additionally, during Aug. 11-12, background value of airborne $\beta_{ext}$ was higher at 4.00 ± 0.71 Mm$^{-1}$ compared to 0.25± 0.11 Mm$^{-1}$ on days without the BB influence, suggesting the additional contribution to $\beta_{ext}$ from the wildfires."***

References

Ely, D. W., et al. (1993). The establishment of the Denver visibility standard. Air And Waste Management Association Annual Meeting, Air And Waste Management Association. **1**.

Groblicki, P. J., et al. (1981). "Visibility-Reducing Species in the Denver "Brown Cloud"- I. Relationships between Extinction and Chemical Composition." Atmos. Env. **15**(12): 2473-2484.

Seinfeld, J. H. and S. N. Pandis (2006). Atmospheric chemistry and physics: from air pollution to climate change. Hoboken, New Jersey, John Wiley and Sons, Inc.

Weber, R. J., et al. (2016). "High aerosol acidity despite declining atmospheric sulfate concentrations over the past 15 years." Nature Geoscience **9**(4): 282-+.

---

## Author Comment (AC2)

We appreciate having received detailed comments from the reviewers. We have revised the manuscript accordingly. Below, you will find our response and the summary of our approach, highlighted in *red,* with modifications to the manuscript highlighted ***in bold***:

*Referee #2:*

Review of Aerosol Optical Extinction during the Front Range Air Pollution and Photochemistry Experiment (FRAPPÉ) 2014 Summertime Field Campaign, Colorado U.S.A.

Overall Comments This paper describes relationships between mass measurements from the mAMS and extinction measurements from a CAPS-PMex instrument in the Front Range of Colorado. The authors derive Mass Extinction Efficiencies for a number of different airmass types (urban, oil and gas, agricultural) based on gas-phase tracer measurements from the study. The paper presents some interesting, relevant results and is appropriate for publication in ACP after some significant revisions described below. The major problems with the submitted manuscript are

1.) The error estimates for the CAPS-PMex are not explained or justified for this study, only a reference is given. The error estimate of 10% seems much too small for small extinction measurements (<10 Mm-1) given that the authors observed significant baseline shifts.

*The frequency distribution plot of the difference in the consecutive baseline values shows that 72% (88%) of the times, baseline shift was within 0.5 $Mm^{-1}$ (1 $Mm^{-1}$). Regardless, after reviewing our procedures for preparing the final version of the CAPS-$PM_{ex}$ data, we realized that we had actually interpolated the baseline values throughout the mission! It was a complete oversight on our part that we didn't explain the procedure for baseline correction accurately in the submitted version. The correct approach is now updated in Section 2.2:* **"Although for majority (72%) of the data, consecutive baseline values had shifted by less than 0.5 $Mm^{-1}$, baseline values were interpolated for a more accurate estimation of optical extinction."**

*The stated 1-s detection limit for CAPS-$PM_{ex}$ was from one of the original papers by Massoli et al. (Massoli et al. 2012). Examining the standard deviations of the measured extinction coefficients during filtered ambient air sampling periods during the project actually indicated a lower 1-s detection limit of ~1.5 $Mm^{-1}$ (assuming 3-$\sigma$). The estimate of the detection limit and the corresponding time are now updated in Section 2.2:* **"The estimated uncertainty in $\beta_{ext}$ is 10%, while the 3-$\sigma$ detection limit for 1-s data under particle free air for the conditions encountered during FRAPPE was ~1.5 $Mm^{-1}$ (Massoli et al. 2010, Petzold et al. 2013)".**

*In general, accuracy of the CAPS-$PM_{ex}$ measurements depends on the accuracy of the baseline and the geometry correction factor (i.e., to know the purge effects). Accuracy in this factor is mostly limited by the accuracy of the CPC during calibrations, which if calibrated carefully with an electrometer, is on the order of ±4%. Considering some uncertainty is also associated with the interpolated baseline values (see our response above regarding baseline correction), we believe an overall 10% uncertainty in the extinction coefficient is conservative, but reasonable.*

2.) The authors see poor correlation between mass measured by the AMS and extinction. This poor correlation is stated to be due to changes in size distribution. However, in some of the plume types, especially urban, there does not appear to be a dramatic difference in the size-distribution of low and high MEE plumes. If the authors are going to conclude that size distribution is responsible for widely varying MEE, this needs to be backed up with some Mie modeling that shows the observed shifts in size

distribution can cause the observed shifts in MEE. It is also possible that varying size cutpoints between the mAMS and CAPS-PMex are significant and this needs to be discussed.
Additional comments are given below.

*In order to determine whether the discrepancy between the aerosol size ranges being sampled by the CAPS-PM$_{ex}$ and AMS had a significant impact on our analysis or not, we have used the size distributions from the Passive Cavity Aerosol Spectrometer Probe (PCASP) instrument on board the C-130 to estimate the ambient scattering coefficients. By using a nominal refractive index of 1.5, estimated scattering (i.e., extinction, while assuming purely scattering aerosols) coefficients were calculated using the measured size distributions up to 800 nm (upper "true" size cut of the AMS) and 2000 nm (upper size cut of the inlet, and thus CAPS-PM$_{ex}$). The slopes of the scatter plots of the estimated scattering coefficients for PM$_{0.8}$ vs. PM$_2$ under the influence of urban, O&G, agricultural, and urban+O&G slopes were 0.95 ± 0.01, 1.0 ± 0.002, and 1.0 ± 0.01, 0.92± 0.01, respectively, indicating that the majority of the signal contribution to extinction originated from aerosols in the size range of the mAMS. We also note that the slope values mentioned above were not highly sensitive to the choice of the refractive index. Changing the refractive index from 1.48 to 1.52 changed the slope values by at most 4%.*

*The following has been added to the text in section 2.2, paragraph 3 to address this issue:*
**"Ambient aerosol size distributions were measured on-board the C-130 by a Passive Cavity Aerosol Spectrometer Probe (PCASP). Estimated extinction values using Mie calculations with a nominal refractive index of 1.5 and the measured PCASP size distributions indicated that particles smaller than 800 nm captured >92% of PM$_2$ extinction values, confirming that the majority of the extinction signal originated from aerosols in the size range of the mAMS. We note that the calculated extinction coefficients were not highly sensitive to the choice of refractive index; only a 4% decrease in the slope of scattering coefficients from PM$_{0.8}$ vs. PM$_2$ was observed by increasing the refractive index from 1.48 to 1.52."**

*Upon closer examination, it became apparent that correlations between β$_{ext}$ and neither of the non-refractory AMS species were high in the urban plumes while as shown in Figure 2, β$_{ext}$ and CO were strongly correlated (r>0.85). We suspect that BC emissions in urban plumes, which are indeed not included in NR-PM$_1$, actually played a role in controlling β$_{ext}$, especially in fresh plumes. It is therefore not surprising that the overall correlation of β$_{ext}$ vs. NR-PM$_1$ in Figure 5a was also not high. The related paragraph in Section 3.2 is now updated to reflect these observations:* **"The scatter plots of dry β$_{ext}$ vs. OA under urban, O&G, and urban + O&G air masses presented correlation coefficients of r = 0.46, 0.72, 0.46, respectively. This observation suggests that O&G emissions are important for organic aerosol contribution to β$_{ext}$. On the other hand, in urban plumes, the correlation between β$_{ext}$ and OA was lower than in O&G plumes while as demonstrated in Figure 2, β$_{ext}$ and CO were strongly correlated under both fresh and aged air masses. These observations suggest that species other than OA, e.g., black carbon, that are co-emitted with CO are also important in driving β$_{ext}$ in urban-influenced air masses."**

Comments on Introduction
1.) It is stated that, "it has been observed that the important contributors to light scattering in the Colorado Rocky Mountains are particulate matter from the urban emissions" Certainly there are dust and smoke events that cause visibility degradation too, in fact the authors show a smoke event. This statement is far too broad.

*This sentence has been modified to express the influence of anthropogenic emissions on visibility:*
**"For example, it has been observed that the important anthropogenic contributors to light scattering in the Colorado Rocky Mountains are particulate matter from the urban emissions (Levin et al. 2009)".**

2.) The authors discuss the wintertime phenomenon of the Denver Brown Cloud but do not discuss summertime visibility issues. Some background on typical extinction or at least PM2.5 mass in the summertime in Denver is needed.

*The last comprehensive aerosol composition measurement study dates back to 1996 while Optical extinction data from the past field studies are not available. After contacting the Colorado Department of Public Health and Environment, we have obtained data on total optical extinction (550 nm) from the transmissiometer measurements in downtown Denver for August 2001-2014. The following section has been added to Introduction to highlight the most recent summertime observations of $PM_{2.5}$ composition and extinction*: **"Previous measurements of summertime particle composition in the Colorado Front Range were conducted during the Northern Front Range Air Quality Study (NFRAQS) between July 17 to August 31, 1996 at several urban and rural sites. The major components of $PM_{2.5}$ mass were identified to be carbonaceous and inorganic aerosols, with carbonaceous aerosols contributing to about 46% of $PM_{2.5}$ mass. The 24-hour average measurements of $PM_{2.5}$ organic carbon, nitrate, and sulfate particles were observed to be 4.2 µg/m$^3$, 0.9-1.2 µg/m$^3$, and 1.4-1.5 µg/m$^3$, respectively (Watson et al. 1998). In response to the wintertime haze episodes observed in the region, the State of Colorado has implemented a visibility standard based on total optical extinction of 76 Mm$^{-1}$ at 550 nm, averaged during a 4-hr period when ambient RH is less than 70% (Ely et al. 1993). Total optical extinction measurements are provided by the Colorado Department of Public Health and Environment's transmissometer installed in Downtown Denver. The average total extinction values of August 2001-2014, ranging from 40-80 Mm$^{-1}$, reveal no significant trend in summertime extinction and visibility in the region since 2001."**

3.) The statement "twofold increase in natural resource extraction" is vague. Is this an increase in wells, in gas production, in oil production? This is relevant because each has a different implication for air quality.

*The twofold increase refers to the number of wells in the regions. We have now added a few sentences in section 1 paragraph 3 of the introduction section, that specifically list the sources of fossil fuel production and various related operations.* **"With a twofold increase in natural resource extraction wells since 2005 to about 24,000 active oil and natural gas (O&G) production wells in 2012, northeastern Colorado has experienced extensive fossil fuel production within the past decade (Scamehorn 2002, Pétron et al. 2014). This includes increases in fossil fuel production from coal bed methane, tight sand and shale natural gas, shale oil, and the associated gases. The emissions from these processes have several environmental impacts such as greenhouse emissions of methane and emissions of non-methane hydrocarbons that impair air quality. Emissions from diesel trucks, drilling rigs, power generators, phase separators, dehydrators, storage tanks, compressors and pipes used in O&G operations also contribute to the regional burden of volatile organic compounds (VOCs), nitrogen oxides, and particulate matter (i.e., black carbon and primary organic carbon) (Gilman et al. 2013)."**

4.) The authors state that emissions from oil and gas include "emissions from industrial equipment known to emit. . .particulate matter." Are they referring to BC from diesel engines? There needs to be clarification here because there are not typically large primary emissions of particulate matter from oil and gas operations.

*As with any off-road diesel equipment, PM emissions from such sources could include BC and POA. This is now clarified as* **"Emissions from diesel trucks, drilling rigs, power generators, phase separators, dehydrators, storage tanks, compressors and pipes used in O&G operations also contribute to the regional burden of volatile organic compounds (VOCs), nitrogen oxides, and particulate matter (i.e., black carbon and primary organic carbon) (Gilman et al. 2013)."**

5.) I don't see a basis for the authors to conclude that photochemical production of ozone is significantly impacted by oil and gas operations. There's a big jump from OH reactivity in the winter to ozone production in the summer and no evidence to back up this logical leap. This also has little to do with the paper, I would suggest removing the last line of page 2 and the first line of page 3.

*The sentence was removed accordingly.*

Comments on Instrumentation and Methodology
1.) The description of the CAPSPMex operating principles is fairly poor. It makes it seem that the extinction is detected by monitoring the phase shift that occurs during 1 transit of the light from the first mirror to the second. This is far from accurate, the entire point of the instrument is to create a very long effective pathlength.

*It was not our intention to suggest such a short path-length of light within the CAPS instrument. Additional information on the CAPS-PM$_{ex}$ effective path length are now included in Section 2.2. of the manuscript.* **"Within the optical cell cavity, the highly reflective mirrors create an effective path length of approximately 2 km. Under the particle free sampling mode, the light emitting diode (LED) light output is directed to the first mirror, while a small fraction goes through the second mirror to the photodiode detector, producing a slightly distorted waveform of the square-wave modulated by the LED, whereas under the aerosol sampling mode, the detector detects a greater distorted waveform, characterized by a phase shift. The vacuum photodiode, which is located behind the second reflective mirror detects and measures that phase shift when the square wave becomes distorted due to interactions with sampled air under a relatively long effective path length."**

2.) Line 4 page 4: An averaging time needs to be given for uncertainty and detection limit statements.

*The stated 1-s detection limit for CAPS-PM$_{ex}$ was from one of the original papers by Massoli et al. (Massoli et al. 2012). Examining the standard deviations of the measured extinction coefficients during filtered ambient air sampling periods during the project actually indicated a lower 1-s detection limit of ~1.5 Mm$^{-1}$ (assuming 3-$\sigma$). The estimate of the detection limit and the corresponding time are now updated in Section 2.2:* **"The estimated uncertainty in $\beta_{ext}$ is 10%, while the 3-$\sigma$ detection limit for 1-s data under particle free air for the conditions encountered during FRAPPE was ~1.5 Mm$^{-1}$ (Massoli et al. 2010, Petzold et al. 2013)".**

3.) Why were baseline values only interpolated when the shift was more than 5 Mm- 1? I see no\reason not to always interpolate them. It also needs to be explained why the baseline would shift that much. Given that most reported measurements are <20 Mm-1, this would seem to be a big issue.

*The following frequency distributions plot of the difference in the consecutive baseline values shows that 72% (88%) of the times, baseline shift was within 0.5 Mm$^{-1}$ (1 Mm$^{-1}$). Regardless, after reviewing our procedures for preparing the final version of the CAPS-PM$_{ex}$ data, we realized that we had actually interpolated the baseline values throughout the mission! It was a complete oversight on our part that we didn't explain the procedure for baseline correction accurately in the submitted version.*

[Figure]

*The correct approach is now updated in Section 2.2: "__Although for majority (72%) of the data, consecutive baseline values had shifted by less than 0.5 Mm-1, baseline values were interpolated for a more accurate estimation of optical extinction.__"*

4.) Error analysis for the CAPS seems to need significantly more attention in general. 10% is clearly not a correct estimate of error in the extinction coefficient when the average extinction is something around 15 Mm-1 and you have a baseline shift of 5 Mm-1. The authors need to do a careful assessment of error in the extinction measurement.

*In general, accuracy of the CAPS-PM$_{ex}$ measurements depends on the accuracy of the baseline and the geometry correction factor (i.e., to know the purge effects). Accuracy in this factor is mostly limited by the accuracy of the CPC during calibrations, which if calibrated carefully with an electrometer, is on the order of ±4%. Considering some uncertainty is also associated with the interpolated baseline values (see our response above regarding baseline correction), we believe an overall 10% uncertainty in the extinction coefficient is conservative, but reasonable.*

5.) Why is particulate nitrate not considered in the NOx/NOy ratio? This seems odd given the authors are measuring particulate nitrate with the mAMS.

*Reviewer has a great point. On average inorganic particulate nitrate was ~9% of total NO$_y$; however, in nitrate-rich plumes, NO$_3^-$ was a more significant fraction. We have therefore included its concentration in the NO$_y$ budget.*

*The following was added to Section 2.2: "__The relationship between the primary emitted nitrogen oxides (NO$_x$) and the higher oxidized species of nitrogen captures the transformation of NO$_x$ in the atmosphere upon aging (Kleinman et al. 2007, Langridge et al. 2012). Thus, measurements of nitric oxide (NO), nitrogen dioxide (NO$_2$), nitric acid (HNO$_3$), particulate phase nitrate (NO$_3^-$) alkyl nitrates (ANs), peroxy acetyl nitrate (PAN), and peroxy propionyl nitrate (PPN) were used to calculate the ratio of primary nitrogen oxides (NO$_x$ = NO + NO$_2$) to NO$_y$ (NO$_y$ = NO$_x$ + HNO$_3$ + NO$_3^-$ + ANs + PAN +PPN) in order to track the extent of photochemical aging in an air mass with non-zero emissions of NO$_x$ (Kleinman et al. 2007, DeCarlo et al. 2010).__"*

6.) A PCASP does not seem to be the best instrument for submicron aerosol analysis. Can the authors comment on why an SMPS or UHSAS was not used in addition to the PCASP? If these instruments were not on the C-130, this needs to be stated.

*Only a nano-SMPS (<100 nm) and PCASP were available among the C-130 payload. Between these two instruments, ambient measurements from PCASP were the more appropriate dataset to use since scattering efficiency for particles smaller than 100 nm at 632 nm is not significant. Additionally, even if an SMPS with a long DMA was available, the relatively long time response of the system for a full scan is not optimum for airborne characterization of pollution plumes. The following statement has been added to the manuscript in section 2.2, paragraph 6:* **"A Passive Cavity Aerosol Spectrometer Probe (PCASP) was available as the only instrument to measure ambient aerosol size distributions in the size range of 0.1-3 μm".** *Note that the SMPS takes 60 seconds per scan, therefore in not well-suited to capture plumes of limited extent or significant spatial variation.*

7.) There needs to be a discussion of the aircraft inlets and size cut points for the various instruments. This is especially critical for the PCASP vs. CAPS-PMex vs. mAMS given that the instruments have different cut points around a micron.

*We have added the following sentence to section 2.2 indicating the size cut of the mAMS:* **"The mAMS inlet was characterized to have a 50% transmission of ~800 nm (physical diameter) particles".** *In order to determine whether the discrepancy between the size ranges being sampled by the CAPS-PM$_{ex}$ and AMS has a significant impact on our analysis or not, we have used the size distributions from the Passive Cavity Aerosol Spectrometer Probe (PCASP) instrument on board the C-130 to estimate the ambient scattering coefficients. By using a nominal refractive index of 1.5, estimated scattering (i.e., extinction, while assuming purely scattering aerosols) coefficients were calculated using the measured size distributions up to 800 nm (upper "true" size cut of the AMS) and 2000 nm (upper size cut of the inlet, and thus CAPS-PM$_{ex}$). The slopes of the scatter plots of the estimated scattering coefficients for PM$_{0.8}$ vs. PM$_2$ under the influence of urban, O&G, agricultural, and urban+O&G slopes were 0.95 ± 0.01, 1.0 ± 0.002, and 1.0 ± 0.01, 0.92± 0.01, respectively, indicating that the majority of the signal contribution to extinction originated from aerosols in the size range of the mAMS. We also note that the slope values mentioned above were not highly sensitive to the choice of the refractive index. Changing the refractive index from 1.48 to 1.52 changed the slope values by at most 4%.*
*The following has been added to the text in section 2.2, paragraph 3 to address this issue:* **"Ambient aerosol size distributions were measured on-board the C-130 by a Passive Cavity Aerosol Spectrometer Probe (PCASP). Estimated extinction values using Mie calculations with a nominal refractive index of 1.5 and the measured PCASP size distributions indicated that particles smaller than 800 nm captured >92% of PM$_2$ extinction values, confirming that the majority of the extinction signal originated from aerosols in the size range of the mAMS. We note that the calculated extinction coefficients were not highly sensitive to the choice of refractive index; only a 4% decrease in the slope of scattering coefficients from PM$_{0.8}$ vs. PM$_2$ was observed by increasing the refractive index from 1.48 to 1.52."**

8.) There needs to be a discussion of how the mAMS data was corrected for collection efficiency, especially given that agricultural plumes had significant nitrate.

*Aerosol concentrations were corrected for vaporizer bounce using composition-dependent CE values. Indeed, CE in plumes heavily impacted by agricultural emissions and thus containing high levels of inorganic aerosol nitrate was higher than the nominal value (0.5).*
*The following sentence has been added to section 2.2 in paragraph 3 for clarification:* **"Aerosol concentrations from the mAMS were corrected for vaporizer bounce using composition-dependent collection efficiencies (Middlebrook 2012)".**

Comments On Results and Discussion
1.) Figure S1 appears to show roads but this is not described in the caption

*Black lines on the map have been updated accordingly as interstates and highways.*

2.) Again, error analysis is a problem. The uncertainty in Bext being stated to be 10% when the authors state that the baseline shifts by over 5 Mm-1 and that the 1-sigma detection limit is 1 Mm-1 seems completely inconsistent. Values of Bext of less than 5 Mm-1 are shown and the authors are suggesting the error is < 0.5 Mm-1 for these measurements but this is half of the 1-sigma detection limit and does not account for baseline shifts. The authors make a major point that they are doing ODR because they want to weight according to the errors in the instruments, but the errors seem incorrect.

*As mentioned above, majority of the baseline shifts were significantly lower than 1 $Mm^{-1}$. Despite that, all baseline values were interpolated for a more accurate determination of the extinction coefficient. The stated 1-s detection limit for CAPS-$PM_{ex}$ was from one of the original papers by Massoli et al. (Massoli et al. 2012). Examining the standard deviation of the measured extinction coefficients during filtered ambient air sampling periods during the project actually indicated a lower 1-s detection limit of ~1.5 $Mm^{-1}$ (assuming 3-$\sigma$). Therefore we believe the uncertainty estimates used as weights of the ODR fit are reasonable. The following text in Section 2.2 has been updated to reflect the DL of this specific instrument "The estimated uncertainty in $\beta_{ext}$ is 10%, while the 3-$\sigma$ detection limit for 1-s data under particle free air for the conditions encountered during FRAPPE was ~1.5 $Mm^{-1}$".*

3.) Page 5 Line 17. The delta ext/co ratios have error bars associated with them, but the authors don't explain what they are. Are these 1-sigma intervals on the slope of the fitted line? For individual measurements the errors would be much larger than these (_11%), I am surprised at how small they are given the poor r of the fits. Also, the authors remove the error in the intercept from the discussion even though this is probably significant.

*The uncertainties in the reported enhancement ratios throughout the paper represent the estimated propagated uncertainties. In Figure 2, the uncertainties are the square-root of the quadratic sum of the relative uncertainties in the ODR fit (1-$\sigma$), CO mixing ratio, and $\beta_{ext}$. The intercept and error has been added to the plots. The following has been added to Section 3.1 to clarify what the uncertainties are: "Uncertainties in the slope values of ODR fits throughout the manuscript represent the estimated propagated uncertainties, in this case, the square-root of the quadratic sum of the relative uncertainties in the ODR fit (1-$\sigma$), CO mixing ratio, and $\beta_{ext}$ coefficient."*
*Note also that Figure 2 is now modified to include data from the August urban plumes only. We had to remove data from July flights due to lack of optimum quantitative quality of CO data during those flights that was reflected on the data archive after the initial submission of the paper. With these changes, indeed correlation coefficients improved from r ~0.6-0.7 to r ~0.85-0.9, confirming that some species that contributed to $\beta_{ext}$ were not co-emitted with CO.*

4.) The authors need to explain how they arrived at the criteria of 0.7, 0.3-0.7, and <0.3 for NOx:NOy ratios.

*Based on the comment of another reviewer, we have limited the data in Figure 2 to only those obtained in urban plumes and during August flights. With the more limited number of data points in these plumes, the aging categories are now redefined with NOx:NOy > 0.5 and <0.5 to represent relatively fresh and aged plumes, respectively.*

5.) Page 5 Line 29. The authors state that the "bulk" of the organic aerosol mass is SOA because the enhancement ratio of organic aerosol increases significantly for aged aerosol. This is not adequate reasoning. The aerosol could be unrelated to CO (from non-combustion sources), as the authors themselves point out. The increasing Bext could be the result of increasing ammonium nitrate when

plumes move farther from the city. The authors need to explain what fraction of aerosol mass is accounted for by the OA:CO enhancement or something in addition to the current argument if they are going to make this sweeping statement. Also, the word "bulk" needs to be quantified in some way.

*We have now limited the data in Figure 2 to those in urban plumes only. In addition, OA fraction in both fresh and aged urban plumes was ~75%; therefore, we don't expect a significant contribution from non-combustion aerosol types to $\beta_{ext}$ data points in Figure 2.The following sentence has been added to Section 3.1 to clarify this: "**The most dominant component of the non-refractory aerosols in urban plumes was OA, with a 74% contribution to NR-PM$_1$ mass. This high OA contribution combined with the observed significant increase in the enhancement ratio of OA to CO with aging (from 0.021 ± 0.009 $\mu$g m$^{-3}$ ppbv$^{-1}$ to 0.11 ± 0.01 $\mu$g m$^{-3}$ ppbv$^{-1}$) suggest that the bulk of the aged urban aerosol mass during the daytime in the Front Range was SOA.  Since $\Delta NO_3^-/\Delta CO$ and $\Delta SO_4^{2-}/\Delta CO$ enhancement ratios did not increase with photochemical aging and demonstrated poor overall correlation coefficients (r <0.35 for $\Delta NO_3^-/\Delta CO$ and r <0.29 for $\Delta SO_4^{2-}/\Delta CO$), the increase in the enhancement ratio of aerosol optical extinction coefficient with CO was likely also driven by SOA formation**."*

6.) Page 6 Line 16. Correlation coefficients of .55 are not "strong"

*The sentence has now been rephrased.*

7.) Page 6 Line 23. Nitric acid + aerosol nitrate is not "total" nitrate, this ignores organic nitrates.

*We have clarified in the text that we refer to inorganic nitrate only since we were considering the partitioning of nitric acid between gas and aerosol phase: "**In fact, the average aerosol inorganic nitrate fraction over total inorganic nitrate (aerosol nitrate/ HNO$_3$ + aerosol nitrate) under agriculture and O&G plumes were 0.25 ± 0.09 and 0.11 ± 0.10, respectively, compared to 0.070 ± 0.071 in urban-influenced plumes.**"*

8.) The relatively weak correlations in urban plumes between Bext and any of the aerosol species (all have r less than 0.55) mass concentrations suggests that there was poor correlation between total AMS mass and Bext. In fact, this is born out in figure 4. This is very confusing and needs to be addressed, especially given the very good correlation between Bext and nitrate in agricultural plumes. The authors suggest it may be caused by a change in size distribution (Figure S2 a-b), but it appears to me that the total mass distributions in these two figures are fairly similar, even though the organic distributions are a little different. Are the authors missing an important aerosol species? Alternatively, is the mismatch in measured aerosol size between the CAPS and AMS significant (Figure S2 suggests this may be the case and that a careful application of the lens transmission is required)? If the authors are going to rest on the size-distribution argument then they need to implement a Mie model of the two distributions shown in S2 and demonstrate that they yield dramatically different Bext for the same total aerosol mass. The size distribution argument for urban + OG is more convincing.

*In order to determine whether the discrepancy between the size ranges being sampled by the CAPS-PM$_{ex}$ and AMS has a significant impact on our analysis or not, we have used the size distributions from the Passive Cavity Aerosol Spectrometer Probe (PCASP) instrument on board the C-130 to estimate the ambient scattering coefficients. By using a nominal refractive index of 1.5, estimated scattering (i.e., extinction, while assuming purely scattering aerosols) coefficients were calculated using the measured size distributions up to 800 nm (upper "true" size cut of the AMS) and 2000 nm (upper size cut of the inlet, and thus CAPS-PM$_{ex}$). The slopes of the scatter plots of the estimated scattering coefficients for*

*$PM_{0.8}$ vs. $PM_2$ under the influence of urban, O&G, agricultural, and urban+O&G slopes were 0.95 ± 0.01, 1.0 ± 0.002, and 1.0 ± 0.01, 0.92± 0.01, respectively, indicating that the majority of the signal contribution to extinction originated from aerosols in the size range of the mAMS. We also note that the slope values mentioned above were not highly sensitive to the choice of the refractive index. Changing the refractive index from 1.48 to 1.52 changed the slope values by at most 4%.*

*The following has been added to the text in section 2.2, paragraph 3 to address this issue:* **"Ambient aerosol size distributions were measured on-board the C-130 by a Passive Cavity Aerosol Spectrometer Probe (PCASP). Estimated extinction values using Mie calculations with a nominal refractive index of 1.5 and the measured PCASP size distributions indicated that particles smaller than 800 nm captured >92% of $PM_2$ extinction values, confirming that the majority of the extinction signal originated from aerosols in the size range of the mAMS. We note that the calculated extinction coefficients were not highly sensitive to the choice of refractive index; only a 4% decrease in the slope of scattering coefficients from $PM_{0.8}$ vs. $PM_2$ was observed by increasing the refractive index from 1.48 to 1.52."**

*Upon closer examination, it became apparent that correlations between $\beta_{ext}$ and neither of the non-refractory AMS species were high in the urban plumes while as shown in Figure 2, $\beta_{ext}$ and CO were strongly correlated (r>0.85). We suspect that BC emissions, which are indeed not included in NR-$PM_1$, in urban plumes actually played an important role in controlling $\beta_{ext}$, especially in fresh plumes. It is therefore not surprising that the overall correlation of $\beta_{ext}$ vs. NR-$PM_1$ in Figure 5a was also not high. The related paragraph in Section 3.2 is now updated to reflect these observations:* **"The scatter plots of dry $\beta_{ext}$ vs. OA under urban, O&G, and urban + O&G air masses presented correlation coefficients of r = 0.46, 0.72, 0.46, respectively. This observation suggests that O&G emissions are important for organic aerosol contribution to $\beta_{ext}$. On the other hand, in urban plumes, the correlation between $\beta_{ext}$ and OA was lower than in O&G plumes while as demonstrated in Figure 2, $\beta_{ext}$ and CO were strongly correlated under both fresh and aged air masses. These observations suggest that species other than OA, e.g., black carbon, that are co-emitted with CO are also important in driving $\beta_{ext}$ in urban-influenced air masses."** *With this explanation, we have deleted sentences in Section 3.3 that emphasized the influence of size distribution shifts on MEE.*

9.) In Figure 4 the removal of the intercept and its associated error is again not mentioned, even though it is a significant issue given the lines don't go through 0,0 and the r values are fairly low. I am again of the opinion that the stated error in the slopes does not accurately represent the error in this slope.

*We appreciate reading reviewer's comment on the intercept in these plots.*

*As mentioned above, the error estimates used as weights in the ODR fits are reasonable. Figure 5 is now updated with the correct weights (taken as the standard deviation of the extinction and aerosol mass) and a more specific mask for urban-influenced plumes. With these changes, the correlation coefficients for all plume types, except urban, have improved. The low correlation of $\beta_{ext}$ and NR-$PM_1$ mass in urban plumes is likely due to contribution of refractory BC that is not accounted for in NR-$PM_1$. To further confirm this, we examined the intercept values of the ODR fits. Except for the agricultural plumes, there appears to be ~2 $Mm^{-1}$ of extinction when NR-$PM_1$ approaches 0, consistent with the hypothesis that there are some refractory components of aerosol that contribute to background extinction. Since PILS data did not indicate large concentrations of $Ca^{2+}$ and $Mg^{2+}$, it's likely that this background extinction is due to background BC.*

*The following sentences have been added to Section 3.3 to explain the observed intercepts of the ODR fits in Figure 5:* **"Based on the values of the intercepts of the ODR fits in Figure 5, it appears that at background levels of NR-$PM_1$ mass, there was a background extinction value of ~2 $Mm^{-1}$ in all, except agricultural, plumes. This observation could be explained by optical extinction due to presence of refractory aerosol species, such as black carbon or dust, which are not accounted for in NR-$PM_1$ mass. High degree of correlation between $\beta_{ext}$ and CO (Figure 2) in urban plumes and low average**

*concentrations of some of the dust components (e.g., calcium and magnesium) support the non-negligible contribution of BC to $\beta_{ext}$ in the Front Range."*

10.) Page 8 Line 7-8. If the authors are going to make this broad conclusion they need to some basic Mie modeling to convince the reader that these shifts in size distribution can generate MEE from 1.8 to 4.1, this does not seem obvious.

*After using the more specific definition of the urban plumes (defined in Section 3.1), the difference in MEE values for urban+O&G plumes vs. other plumes is not significant considering the uncertainties in the slopes of the ODR fits. We have therefore removed the reference to the importance of shifts in the size distributions and Figure 7.*

11.) In Figure 7a "total mass" needs to be changed to "total non-refractory PM-1 mass", though even the PM-1 designation is questionable with AMS data.

*The reviewer is correct in mentioning that the inlet system of the AMS is not exactly a $PM_1$ inlet. Transmission efficiency tests of the PCI and the lens system on this specific mAMS indicated a 50% transmission of particles with physical diameter ~ 800 nm, which is comparable to other AMSs. (e.g., Bahreini et al, AST, 2008). For consistency with the common notation for AMS measurements, we have updated the label on the plot as "Total non-refractory $PM_1$ mass".*

12.) In Figure 7a the fit to the non-biomass burning emissions does not seem to track the data. The authors need to discuss this. Perhaps it is due to including moderate HCN data points.

*We have updated this Figure to include data with masks designating the 4 plume types that have been examined in detail in the paper. This has greatly improved the correlations of $\beta_{ext}$ and NR-$PM_1$ mass, with r values ~0.75-0.80 as well as the trends of the weighted ODR fits. Please note that because BB also contributes to atmospheric CO, we decided the conclusions drawn from the scatter plot of $\beta_{ext}$ vs. CO in the presence and absence of BB could not be as robust as desired and have therefore deleted panel b.*

13.) Again in Figure 7 the intercepts are removed and not discussed, they need to be discussed because they are very different between biomass burning and non-biomass burning and this has physical meaning.

*Although previously the intercepts were not indicated in this figure, a discussion was provided in the text to highlight the significance of the higher intercepts observed during biomass burning days. Data in Figure 7 are now limited to those sampled in urban, urban + OG, OG, and agricultural plumes, with a more specific definition of urban plumes, and the following text in Section 3.4 has been updated accordingly:* **"MEE values were analyzed for days with and without the BB influence, using weighted linear ODR fit analysis, as explained previously. As seen in Figure 7, average MEE on Aug. 11-12 was ~63% greater compared to days without the influence of BB (3.65±1.16 $m^2$/g vs. 2.24±0.71 $m^2$/g). Additionally, during Aug. 11-12, background value of airborne $\beta_{ext}$ was higher at 4.00 ± 0.71 $Mm^{-1}$ compared to 0.25± 0.11 $Mm^{-1}$ on days without the BB influence, suggesting the additional contribution to $\beta_{ext}$ from the wildfires."**

References

DeCarlo, P. F., et al. (2010). "Investigation of the sources and processing of organic aerosol over the Central Mexican Plateau from aircraft measurements during MILAGRO." Atmospheric Chemistry and Physics **10**(12): 5257-5280.

Ely, D. W., et al. (1993). The establishment of the Denver visibility standard. Air And Waste Management Association Annual Meeting, Air And Waste Management Association. **1**.

Gilman, J. B., et al. (2013). "Source Signature of Volatile Organic Compounds from Oil and Natural Gas Operations in Northeastern Colorado." Environmental Science & Technology **47** 1297–1305.

Kleinman, L. I., et al. (2007). "Aircraft observations of aerosol composition and ageing in New England and Mid-Atlantic States during the summer 2002 New England Air Quality Study field campaign." Journal of Geophysical Research-Atmospheres **112**(D9).

Langridge, J. M., et al. (2012). "Evolution of aerosol properties impacting visibility and direct climate forcing in an ammonia-rich urban environment." Journal of Geophysical Research: Atmospheres **117**(D21): n/a-n/a.

Levin, E. J. T., et al. (2009). "Aerosol physical, chemical and optical properties during the Rocky Mountain Airborne Nitrogen and Sulfur study." Atmospheric Environment **43**(11): 1932-1939.

Massoli, P., et al. (2010). "Aerosol Light Extinction Measurements by Cavity Attenuated Phase Shift (CAPS) Spectroscopy: Laboratory Validation and Field Deployment of a Compact Aerosol Particle Extinction Monitor." Aerosol Science and Technology **44**(6): 428-435.

Massoli, P., et al. (2012). "Aerosol Light Extinction Measurements by Cavity Attenuated Phase Shift (CAPS) Spectroscopy: Laboratory Validation and Field Deployment of a Compact Aerosol Particle Extinction Monitor." Aerosol Sci. Technol. **44**: 428-435.

Middlebrook, A., et al. (2012). "Evaluation of Composition-Dependent Collection Efficiencies for the Aerodyne Aerosol Mass Spectrometer using Field Data." Aerosol Science and Technology **46**: 258–271.

Pétron, G., et al. (2014). "A New Look at Methane and Nonmethane Hydrocarbon Emissions from Oil and Natural Gas Operations in the Colorado Denver-Julesburg Basin." Journal of Geophysical Research-Atmospheres **119**(11): 6836-6852.

Petzold, A., et al. (2013). "Intercomparison of a Cavity Attenuated Phase Shift-based extinction monitor (CAPS PMex) with an integrating nephelometer and a filter-based absorption monitor." Atmospheric Measurement Techniques **6**(5): 1141-1151.

Scamehorn, H. L. (2002). High Altitude Energy: A History of Fossil Fuels in Colorado. Boulder, CO. Colorado, University Press of Colorado.

Watson, J. G., et al. (1998). Northern Front Range Air Quality Study Final Report, Colorado State University.

---

## Author Response (AR2)

Comments to the Author:

Dear Authors, you've clearly taken into account both reviewer comments/suggestions and have revised the manuscript accordingly. Thus, I gladly accept this paper for ACP. However, I have only one question that remains from the review process. I agree with Reviewer 1, why wasn't PMF considered for the organics and possibly applied in this analyses presented here? I think the authors at least need to acknowledge this in the main text; specifically, that PMF could be applied in future work if the authors decide not to do that here. Other than this minor issue, this will be a nice contribution to ACP.

Best wishes to you all,

Jason Surratt

*Our Response:*

*Dear Dr. Surratt,*

*We appreciate your quick review of our revised manuscript. We have tentative plans to carry out PMF analysis on the summertime OA spectra, compare the results with those obtained at Erie tower during winter 2011, and present the analysis in a separate paper. We have now added the following sentence in Section 3.1 to address your concern: "**Although carrying out positive matrix factorization analysis on the measured OA mass spectra during FRAPPÉ is beyond the scope of this paper, such analysis in the future would be conducive in confirming the large contribution of SOA to the measured OA**."*

*We look forward to hearing from you.*

*Regards,*

*Roya Bahreini*

[revised manuscript text omitted]